# Divergent roles of FT-like 9 in flowering transition under different day lengths in *Brachypodium distachyon*

Zhengrui Qin[1], Yuxue Bai[1], Sajid Muhammad[1], Xia Wu[1], Pingchuan Deng[1], Jiajie Wu[2], Hailong An[2] & Liang Wu [1]

Timing of reproductive transition is precisely modulated by environmental cues in flowering plants. Facultative long-day plants, including Arabidopsis and temperate grasses, trigger rapid flowering in long-day conditions (LDs) and delay flowering under short-day conditions (SDs). Here, we characterize a SD-induced FLOWERING LOCUS T ortholog, FT-like 9 (FTL9), that promotes flowering in SDs but inhibits flowering in LDs in *Brachypodium distachyon*. Mechanistically, like photoperiod-inductive FT1, FTL9 can interact with FD1 to form a flowering activation complex (FAC), but the floral initiation efficiency of FTL9-FAC is much lower than that of FT1-FAC, thereby resulting in a positive role for FTL9 in promoting floral transition when FT1 is not expressed, but a dominant-negative role when FT1 accumulates significantly. We also find that CONSTANS 1 (CO1) can suppress *FTL9* in addition to stimulate *FT1* to enhance accelerated flowering under LDs. Our findings on the antagonistic functions of FTL9 under different day-length environments will contribute to understanding the multifaceted roles of FT in fine-tune modulation of photoperiodic flowering in plants.

[1] College of Agriculture and Biotechnology, Zhejiang University, 310058 Hangzhou, China. [2] State Key Laboratory of Crop Biology, Shandong Agricultural University, 271018 Tai'an, China. These authors contributed equally: Zhengrui Qin, Yuxue Bai. Correspondence and requests for materials should be addressed to L.W. (email: liangwu@zju.edu.cn)

The correct flowering timing is a critical determinant of the adaptation to different environments in plants. Flower meristem genes would be timely induced by a flowering activation complex (FAC), constituted by FLOWERING LOCUS T (FT), a small phosphatidylethanolamine binding protein (PEBP), and FD, a bZIP transcription factor by an assistance of scaffold protein 14-3-3 s in the cell nucleus, when plants respond to favorable flowering environments[1,2]. The diversification of FTs is important for floral initiation in plants, as different FTs or splicing isoforms may play opposite roles in flowering onset[3,4].

Monitoring day length is one of the most important environmental cue for plant decision of reproductive transition. Facultative long-day plants, including *Arabidopsis thaliana*, trigger rapid flowering in long-day conditions (LDs) while delay flowering under short-day conditions (SDs)[5]. Under LDs, *FT* mRNA is induced by a key transcriptional regulator, CONSTANS (CO), though a direct interaction or an indirect association with *FT cis*-elements[6–8]. Although clock-regulated and light-dependent CO mRNA and protein accumulation in plants are conserved and crucial for sensing day-length changes to regulate *FT*, however, lineage-specific plants may evolve special factors to control *FTs* under different light conditions[9,10]. For example, a temperate grasses-specific microRNA, miR5200, have been recently illustrated in a photoperiodic control of two *FTs* (*FT1* and *FT2*) through a mRNA degradation approach[11–13].

Compared with LD-induced floral promotion that has been extensively explored, the nature of default floral induction under SDs in LD-plants remains enigmatic[14]. Flowering of LD-plants under SD conditions has been considered mainly through plant hormone gibberellin (GA) signaling, but how plants percept and deliver GA signaling to trigger floral onset are still controversial[15]. Thereby, identification of more modulators in non-inductive environments can help understanding the exact molecular pathway of flowering initiation in plants.

Wheat and barley, two temperate grasses, provide us major sources of food and energy worldwide. Photoperiod sensitivity is an important agronomic event of wheat crop. Genetic study reveals that the adaptation to photoperiod in barley and wheat is predominantly determined by two Quantitative Trait Loci, *PHOTOPERIOD 1 and 2* (*Ppd-1* and *Ppd-2*)[16]. Ppd-1, an ortholog of *A. thaliana* PSEUDO-RESPONSE REGULATOR protein in circadian signaling, plays a major role in ear emergence in LDs through activation of *FT1* transcription[17]. *Ppd-2* was identified to produce flowering time variation in SDs over two decades ago, however, the gene encoded Ppd-2 has not been clearly confirmed to date[18]. A short-day induced *FT*, *FT3*, is located to the same region in the chromosome as *Ppd-2* in barley, thereby recently thought to be the candidate encoder of Ppd-2, but the biological manner of FT3 is still unknown[19–21]. There is a close homolog named FT5 within the FT3 clade in barley, but its expression pattern and physiological involvement have not been determined to date as well[19,22]. Thereby, further molecular characterization of FT3 and FT5 is important to breed ideal genotypes for wider cultivation, using different combinations of their alleles in wheat and barley.

*Brachypodium distachyon* is an ideal model temperate grass for cereal flowering machinery investigation, due to its simple growth condition that can be artificially controlled with light and temperature[23]. *B. distachyon* harbors all orthologs of the three core flowering proteins VERNALIZATION 1 (VRN1), VRN2, and FT1 in barley and wheat[24]. As those in barley and wheat, FT1 and VRN1 are behaved as promoters of flowering by forming a positive feedback loop in *B. distachyon*[11,24,25]. However, *VRN2* in *B. distachyon* has been found induced during the vernalization process, in contrast with that in wheat and barley, which is suppressed by a cold treatment[26,27]. FT2 is a close paralog of FT1

in the temperate grasses. Recently, overexpression of *FT2* has been observed giving rise to a precocious flowering both in *B. distachyon* and barely[28], suggesting a similar role of FT2 in timing of the transition in *B. distachyon* and other temperate cereals. Accordingly, these results indicate that the orthologs of flowering proteins may have either similar or distinct functions in *B. distachyon*, wheat and barley.

There are also two protein members (FTL9 and FTL10) in FT3 clade in *B. distachyon*. Unlike that of in barley, FTL9 in *B. distachyon* is not affiliated in the same phylogenetic subgroup as FTL10 (Supplementary Fig. 1a), implicating that FTL9 is not derived from a recent duplication of FTL10 in *B. distachyon*, and thus may evolve alternative roles in flowering control[29]. In this study, we characterized FTL9 in *B. distachyon*, and observed a striking flowering performance in *FTL9* loss-of-function and gain-of function plants. Mechanistically, we found that FACs by FTL9 and by canonical FT (FT1) had different flowering inductive abilities, resulting in an opposite behavior of FTL9 in flowering control in different day lengths. Moreover, we revealed that CO1 could suppress *FTL9* to promote flowering under inductive LD conditions. Taken together, we illustrate a photoperiodic flowering pathway that is mediated by FTL9 and CO1 in *B. distachyon*.

## Results

**FTL9 is highly accumulated under SDs.** In contrast to two FT-like proteins in *A. thaliana*, there are 6 FT orthologs in temperate grasses[3,20] (Supplementary Fig. 1a). To explore the photoperiodic responses of *FT* orthologs in temperate grasses, we examined the expressions of all *FT*-like genes at Zeitgeber time 4 (ZT4) under SDs and LDs in *B. distachyon*. *FT1, FT2, FTL10*, and *FT4* are highly expressed after dawn under LDs whereas repressed in SDs (Fig. 1a). *FT6* is difficult to be detected at any day-length conditions, perhaps due to its low expression (Fig. 1a). *FTL9* is the sole *FT* that is highly accumulated in SDs, while dramatically repressed in LDs (Fig. 1a), reminiscent that of *FT3* in barley and wheat (Supplementary Fig. 1b)[19–21], indicating that *FTL9* in *B. distachyon* is a specific short-day induced *FT*. Thus, we further investigated its biological significance and molecular basis in detail.

To further pursue the effects of SDs on *FTL9* expression, we examined the dynamic changes of *FTL9* when plants were shifted from different day-length environments. There was a significant reduction of *FTL9* transcription after plants moved from SDs to LDs for 3 days, and almost disappeared after 14-days movement (Fig. 1b). However, when LD-grown plants were transferred to SDs for 7 days, more than 40-fold *FTL9* could be induced compared with that of under consistent LDs (Fig. 1c). Since *FT* family genes usually have diurnal expression rhythms, we examined the oscillation of *FT1* and *FTL9* every 4 h during the day-night cycle. In line with our previous results[11], *FT1* displayed an expression peak at 4 h before dawn when plants were grown under LDs, but fully repressed under SDs (Supplementary Fig. 2a). By contrast, *FTL9* was expressed to the highest level at the end of dark when plants were grown in SDs, and it was significantly repressed at any time points when plants were grown in LDs (Supplementary Fig. 2b).

PHYTOCHROME C (PHYC) is the critical light sensor in temperate grasses, and a couple of key flowering genes including *FT1* and *CO1* are greatly attenuated in *phyc-1* mutant in LD photoperiod (Supplementary Fig. 3)[30–32]. By contrast, we found that there was an increase of 300-fold *FTL9* in *phyc-1* mutant than in wild-type plants under LDs (Fig. 1d), but only a very mild increase under SDs (Fig. 1e), suggesting that *FTL9* is strictly controlled by PHYC under inductive photoperiod.

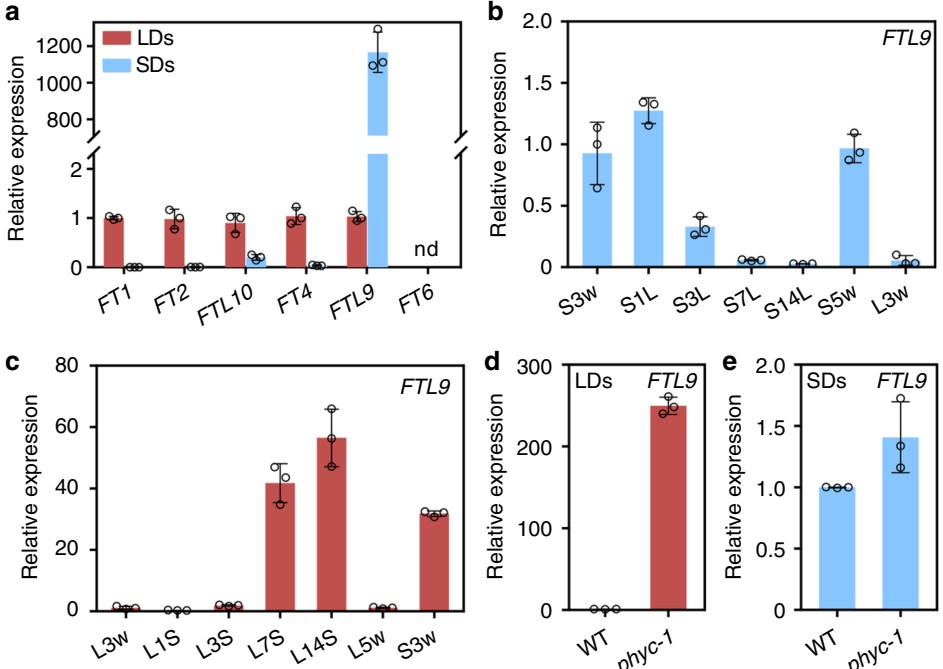

**Fig. 1** FTL9 is specifically induced under SDs in *B. distachyon*. **a** qRT-PCR analysis of six *FT* orthologous gene expressions at ZT4 under SDs and LDs. *UBC18* was used as an internal control for normalization of qRT-PCR results. Plant newly expanded leaves were harvested for RNA isolation and gene expression analysis. Error bars indicate s.d. **b** Time course analysis of *FTL9* expression at ZT4 when three-week-old SD plants were shifted to LDs for 1 day (S1L), 3 days (S3L), 7 days (S7L), and 14 days (S14L), respectively. S3w and S5w indicate that *B. distachyon* were grown under consistent SDs for 3 and 5 weeks. L3w indicates that *B. distachyon* were grown under LDs for 3 weeks as a control. *UBC18* was used as an internal control for normalization of qRT-PCR results. Error bars represent s.d. **c** Time course analysis of *FTL9* expression at ZT4 when three-week-old LD plants were shifted to SDs for 1 day (L1S), 3 days (L3S), 7 days (L7S) and 14 days (L14S), respectively. L3w and L5w indicate that *B. distachyon* were grown under consistent LDs for 3 and 5 weeks. S3w indicates that *B. distachyon* were grown under SDs for 3 weeks as a control. Error bars represent s.d. **d** qRT-PCR analysis of *FTL9* expression under LDs in the wild-type Bd21-3 and *phyc-1* homozyous mutants. *UBC18* was used as an internal control for normalization of qRT-PCR results. Error bar means s.d. **e** qRT-PCR analysis of *FTL9* expression under SDs in the wild-type Bd21-3 and *phyc-1* homozyous mutants. Error bar means s.d.

Collectively, these results indicate a significant induction of *FTL9* by short days in *B. distachyon*.

**FTL9 is involved in flowering in SDs.** FAC formation mediated by FT is essential for floral initiation. Our yeast two-hybrid, bimolecular fluorescence complementation (BiFC) and firefly luciferase complementation imaging (FLCI) assays show that FTL9 can associate with FD1 and 14-3-3 proteins to constitute a functional FAC as FT1, a canonical LD-induced florigen encoder (Fig. 2a–d and Supplementary Fig. 4)[3].

Subsequently, we asked how FTL9-mediated FAC controls flowering in *B. distachyon*. To address this question, we generated *FTL9* loss-of-function mutants through a CRISPR-cas9 approach (Fig. 2e). Since *FTL9* is specially accumulated in SDs, we determined flowering time in two homozygous frame-shift mutants of *ftl9* under SDs. We observed that they flowered almost 190 days after grown in soil under SDs, two month later than the wild-type (WT) plants (Fig. 2f, g). Together with the significant repressions of *VRN1* and *FUL2*, two flowering downstream genes, in *ftl9* mutants (Fig. 2h, i), these findings suggest that FTL9 is involved in flowering control under SD photoperiod.

**FTL9-FAC has a mild flowering-inductive effect under SDs.** To further explore the functions of FTL9 in SD-mediate flowering, we overproduced FTL9 in *B. distachyon*. *FTL9* transcript level was 10-fold to 20-fold increased in our several independent *FTL9* over-expression (*FTL9-OE*) lines (Supplementary Fig. 5). As expected, the heading-date of *FTL9-OE* plants was significantly shortened

under SDs (Fig. 2j, k), and *VRN1* and *FUL2* expressions were dramatically increased compared with WT (Fig. 2l, m), suggesting that FTL9 can indeed promote heading in *B. distachyon* under SDs. Nevertheless, it is notable that the reduction of flowering time in *FTL9-OE* is far from that in *FT1* over-expression plants, which can lead to an extremely early flowering[11,23–25], implying that although FTL9 is a positive regulator of flowering under SDs, its flowering inductive activity may not be dramatic. Interestingly, the expressions of *FT1* are similar in *FTL9-OE* lines and wild-type plants (Supplementary Fig. 6), implicating that the induction of flowering under SDs by FTL9 is neither by a direct nor an indirect trigger of *FT1*.

We further investigated why FTL9 had such a mild flowering inductive effects under SDs. Since FAC in temperate grasses have been demonstrated with an ability to bind *VRN1* cis-elements and promote *VRN1* expression in wheat[2,33], we determined the binding capacity of FTL9-FD1 and FT1-FD1 module to *VRN1* promoter (VRN1p) when they simultaneously introduced into *Nicotiana benthamiana* leaf cells. As shown in Fig. 3a, the signal of luciferase (LUC) reporter triggered by FTL9-FAC-VRN1p was much lower than that by FT1-FAC-VRN1p, suggesting a finitely propulsive role of FTL9 in plant flowering.

To assess what makes the difference of flowering inductive capacity of FT1 and FTL9 in *B. distachyon*, we artificially swapped FTL9 and FT1 PEBP domain and transiently introduced them with FD1 into *N. benthamiana* leaf cells to compare the VRN1p-LUC signal. We found that the VRN1p-LUC activation by FTL9 could be obviously enhanced when the FTL9 PEBP domain was changed to that of FT1, whereas VRN1p-LUC activation by FT1

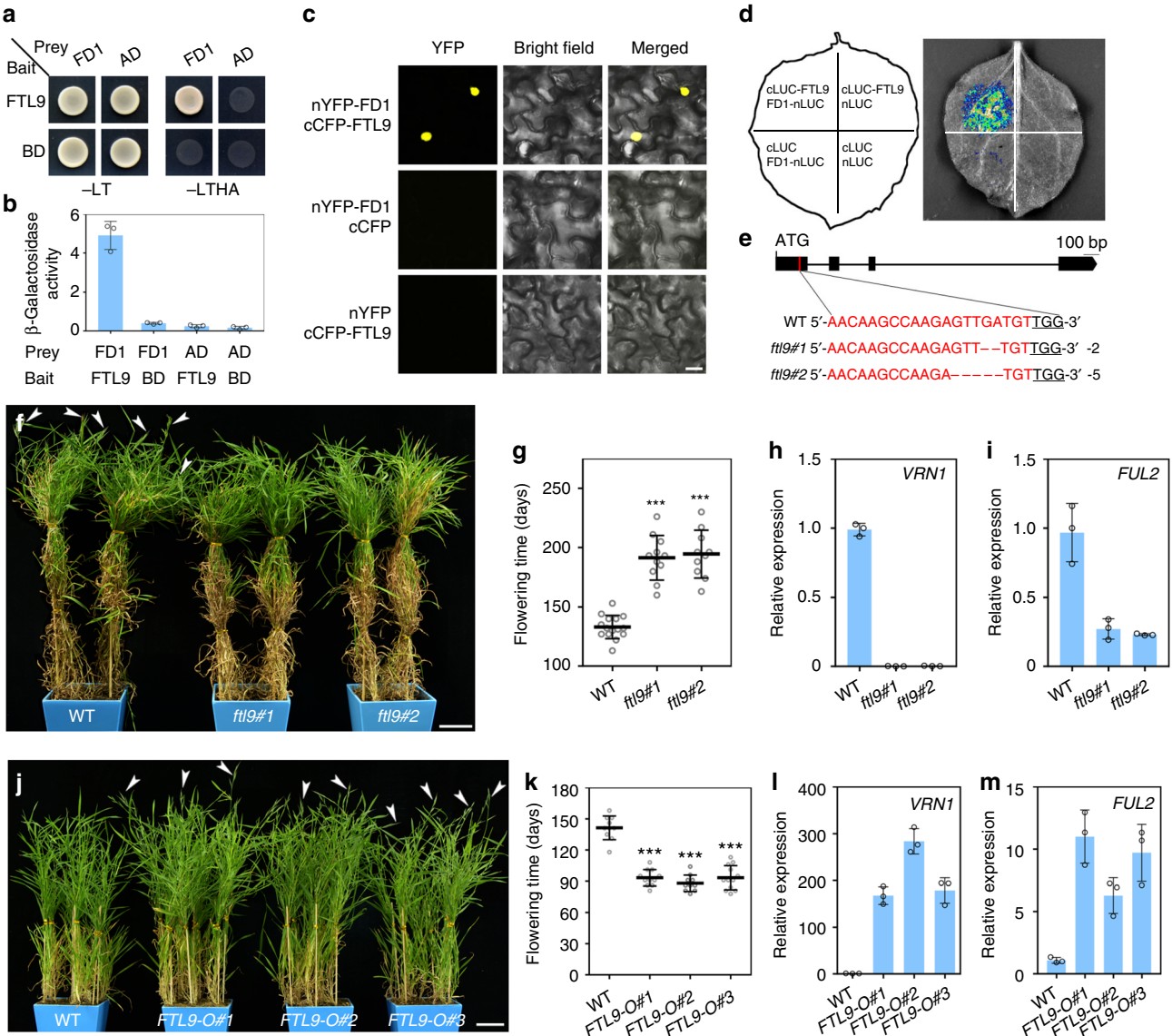

**Fig. 2** FTL9 is a promoter of flowering under SDs in *B. distachyon*. **a** Interaction analysis between FTL9 and FD1 in yeast cells. Yeast cells growing on selective media without Leu, Trp, His and Ade (-LTHA) represent positive interactions. **b** Relative quantification of FTL9 and FD1 interactions by β-Gal activity in yeast two-hybrid assays. Bars indicate s.d. **c** BiFC assays for determination of FTL9 and FD1 interactions in *N. benthamiana* leaf cells. Scale bar, 20 μm. **d** Firefly luciferase complementation imaging (FLCI) assays of FTL9 and FD1 interactions in *N. benthamiana* leaves. **e** CRISPR/Cas9-mediated target mutagenesis of *FTL9*. Top: a schematic diagram of the *FTL9* bearing the CRISPR-Cas9 target sites. Exons and introns are represented by black boxes and lines, respectively. The lower sequences show the alignment of wild-type Bd21, *ftl9-1* and *ftl9-2* sequences containing the CRISPR-Cas9 target sites. The *ftl9-1* and *ftl9-2* mutants contain a 2-bp and 5-bp deletion, respectively. The black underlined sequences mean protospacer adjacent motifs.
**f** Representative phenotypes of two *ftl9* mutants under SDs. White arrows point to spikes. Scale bar, 5 cm. **g** Flowering time of wild-type (WT) Bd21 and the *ftl9* mutants under SDs. At least 10 plants for each line were scored. (Student's *t* test, ***P < 0.001). **h** qPCR analysis of flowering downstream gene *VRN1* expression in eight-week-old wild-type and the indicated *ftl9* mutants under SDs. Error bars indicate s.d. **i** qPCR analysis of flowering downstream gene *FUL2* expression in eight-week-old wild-type and two *ftl9* mutants under SDs. Error bars indicate s.d. **j** Representative phenotypes of three *FTL9* over-expressing plants under SDs. White arrows point to spikes. Scale bar, 5 cm. **k** Flowering time of wild-type Bd21 and the *FTL9-OE* plants under SDs. At least 10 plants for each line were scored. (Student's *t* test, ***P < 0.001). **l** qPCR analysis of flowering downstream gene *VRN1* expression in eight-week-old wild-type and the *FTL9-OE* plants under SDs. Error bars indicate s.d. **m** qPCR analysis of flowering downstream gene *FUL2* expression in eight-week-old wild-type and the *FTL9-OE* plants under SDs. Error bars indicate s.d.

could be significantly compromised when the FT1 PEBP domain was changed to that of FTL9 (Fig. 3b, c), suggesting that the distinction of PEBP domain of FTL9 and FT1 is an important determinant of their divergent flowering initiation activity.

Subsequently, we attempted to explore which amino acid confers the specific flowering activity of FTL9 compared with FT1, and thereby compared the proteins of FTL9 with other FTs that are known with a high flowering initiation activity in long-

day plant *B. distachyon* and *A. thaliana*, including BdFT1, BdFT2, BdFTL10, AtFT and At TWIN SISTER OF FT (AtTSF) (Supplementary Fig. 7)[11,25,34,35].We assumed that the specific amino acids of FTL9 at PEBP domain that are completely different with those in other FTs might be responsible for the FTL9 mild flowering activity (Supplementary Fig. 7). However, when we swapped Lysine (Lys, K) at residue 128, the sole amino acid that is entirely distinct from that in other FTs at FTL9 PEBP

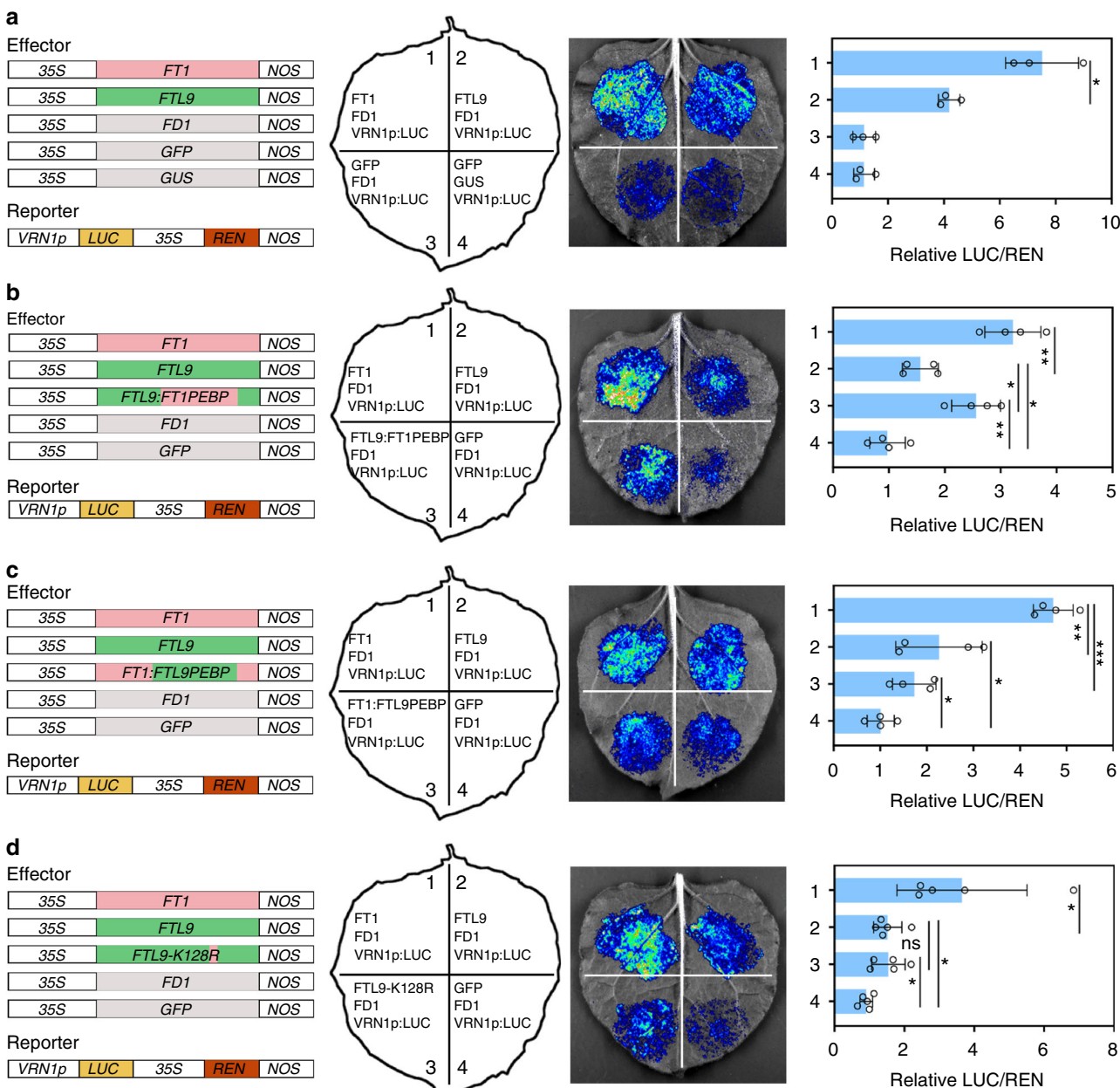

**Fig. 3** FTL9-FAC has a weak flowering inductivity in *B. distachyon*. **a** The flowering inductive efficiency of FTL9-FAC is lower than that of FT1-FAC. Error bars indicate s.d. from different injected leaves. **b** FAC activity by FTL9 can be enhanced when the FTL9 PEBP domain is changed to that of FT1. PEBP domain of FTL9 is artificially swapped to that of FT1 and introduced to *A.tumefaciens* with FD1 and 3 kb VRN1 promoter-mediated LUC constructs, and co-infiltrated into *N. benthamiana* leaves. Error bars indicate s.d. **c** FAC activity by FT1 can be compromised when the FT1 PEBP domain is changed to that of FTL9. PEBP domain of FT1 is artificially swapped to that of FTL9 and introduced to *A.tumefaciens* with FD1 and 3 kb VRN1 promoter-mediated LUC constructs, and co-infiltrated into *N. benthamiana* leaves. Error bars indicate s.d. **d** Lysine at residue 128 of FTL9 is not responsible for the specific flowering ability of FTL9 in *B. distachyon*. Error bars indicate SD. (Student's *t* test, *$P < 0.05$, **$P < 0.01$, ***$P < 0.001$) Left: Different combinations of effectors and the reporter transiently introduced in *N. benthamiana* leaves. Middle: Representative photograph of firefly luciferase fluorescence signals when the indicated reporter and effectors were introduced in *N. benthamiana* leaves. Right: Relative reporter activity (LUC/REN) in *N. benthamiana* leaves expressing the indicated reporters and effectors

domain, to Arginine (Arg, R), we did not observe a conspicuous alteration of VRN1p-LUC signal compared with the original FTL9 (Fig. 3d), suggesting that only this Lys cannot be responsible for the specific flowering initiation capacity of FTL9, and whether a large fragment or other amino acids in FTL9 are the causes of its particular flowering action requires further determined.

**Over-expression of *FTL9* leads to a flowering delay under LDs**. Because the flowering acceleration in *FTL9-OE* plants under SDs is not remarkable as that in canonical FT overproducing plants, we wondered what's their performance under LDs. Strikingly, we observed a significant delay rather than a precocity in heading in *FTL9-OE* plants (Fig. 4a, b), implicating that FTL9 may interfere plant flowering under LD conditions. Our further examination of

*VRN1* expression confirmed that over-expression of *FTL9* can inhibit flowering under inductive photoperiod environments (Fig. 4c).

Next, we wondered why FTL9 has completely opposite actions in flowering control under different photoperiods. As we described above, FTL9 can interact with FD1 to form a functional FAC, but its flowering induction is not as efficient as FT1, thus it is possible that FTL9 can attenuate the FT1-FAC activity by competing with FT1 to FD1 when FT1 exists. To test this hypothesis, we transiently co-expressed FTL9, FT1, and FD1 in *N. benthamiana* leaves, and determined their effects on binding to *VRN1* promoter. As shown in Fig. 4d, the LUC signal triggered by the association between FT1-FAC and VRN1p is significantly compromised when FTL9 is introduced, however, there is no any decreases of LUC intensity when a control GFP or FT1 are co-expressed, suggesting that FTL9 is conferred with an inhibitive ability to FT1-FAC (Supplementary Fig. 8).

Taken together, the lower efficiency of FTL9-FAC than FT1-FAC in floral initiation leads to a positive role of FTL9 in plant reproductive transition under SDs when FT1 is not expressed, while a dominant-negative effect when FT1 is significantly accumulated under LDs.

**CO1 is a repressor of FTL9**. CO has been shown as the central photoperiodic sensor that triggers flowering through activation of florigen genes in *A. thaliana* and many other plants[36,37]. Due to the contrary actions of *FTL9* in flowering under different day lengths, we are curious whether *FTL9* can be controlled by CO-like proteins in *B. distachyon*.

There are two homologs of CO proteins in *B. distachyon*, termed CO1 and CO2, respectively (Supplementary Fig. 9a). To specifically knock down *CO1* in *B. distachyon*, we chose the specific region of *CO1* to produce double-strand RNA to silence *CO1* in transgenic plants. As expected, *CO2* expression is not decreased in *CO1-RNAi* lines (Supplementary Fig. 9b, c). Moreover, we generated *CO1* over-expression lines (Supplementary Fig. 9d), and demonstrated that CO1 in *B. distachyon* could be functioning as a general CO-like protein, which promotes flowering by activating *FT1*, and thereby increasing downstream *VRN1* expression (Supplementary Figs. 10 and 11). Interestingly, when we measured the expression of *FTL9* in *CO1-RNAi* plants under LDs, we found that *FTL9* was obviously increased compared with WT (Fig. 5a). Furthermore, we observed a dramatic suppression of *FTL9* in *CO1-OE* transgenic lines under SD conditions (Fig. 5b), suggesting that *FTL9* is able to be repressed by CO1.

Consistent with genetic data, our molecular analysis found that the LUC signal triggered by *FTL9* promoter in *N. benthamiana* leaves was pronouncedly decreased when we transiently introduced the CO1 protein (Fig. 5c). These data indicates that CO1 regulates flowering in *B. distachyon* through *FTL9* repression in addition to *FT1* activation.

## Discussion
Day length is a critical seasonal indicator that influences plant reproductive success. Current understanding of photoperiodic flowering in LD plants mostly be restricted to a long-day accumulated FT, which is temporal and spatial promoted by CO, a oscillating sensor whose accumulation occurs at the right time of the day. This model nicely explains how flowering is induced under LDs, however, how plants trigger SD flowering is puzzled. In this study, we characterized a non-inductive short-day induced FT, FTL9, in *B. distachyon* (Fig. 5d). On the one hand, the mild flowering promotive activity of FTL9 is to ensure plant flowering eventually to propagate next generation under SDs; on the other

hand, due to the weaker floral induction activity of FTL9-FAC compared with that of FT1-FAC, FTL9 becomes a dominate-negative repressor of flowering when FT1 is significantly accumulated under LDs.

The action of FTL9 is particular of biological significance, especially for winter varieties of temperate grasses, which must go through a short-day and low-temperature winter to confer flowering inductive ability. The loss-of-function of *FTL9* can give rise to a delayed flowering under SD photoperiod, otherwise, the sensitive floral organs would be subject to a low-temperature damage if the plants are premature flowering in winter. When day length shifts to a long period in spring, winter plants would initiate flowering through a combinative approach, including activation of *FT1* and decrease of *FTL9*. Thereby, our mechanistic elucidation of divergent roles of FTL9 under different photoperiods can provide some insights into creating proper cereal accessions in adaptation to broader agricultural environments and climate changes.

It is worthwhile to note that although *VRN1* is increased in *FTL9-OE* plants whereas decreased in *ftl9* mutants, the acceleration of flowering by FTL9 in SDs may not be solely attributed to VRN1 activation, because it has been shown that over-expression of *VRN1* does not result in a rapid flowering under SDs[26]. Besides, even *VRN1* is higher in *enhancer of zeste-like*, a mutant of orthologous gene of *A. thaliana CURLY LEAF 1*, than wild-type plants under SDs, it still cannot trigger a more rapid flowering[38]. Therefore, expression of *VRN1* may not be sufficient to induce flowering in *B. distachyon* under SD. Further identification of more *FTL9* regulatory genes is valuable to illustrate the cause of reproductive transition in *B. distachyon* under non-inductive environments.

A recent study showed that most vernalization varieties of barley possessed the *ppd-H2* allele, a recessive allele of *FT3*[39]. There are two close homologs of *FT3* in barley, named FT3 and FT5, and perhaps they are evolved from a gene duplicate event[22]. Although *B. distachyon* FTL9 and FTL10 belong to a big clade of FT3 proteins in temperate cereals, FTL9 in *B. distachyon* is not affiliated to the same subclade with FT3 in barley (Supplementary Fig. 1a). Because FT3 in barley accelerates the initiation of spikelet primordia independently of the photoperiod[34], while FTL9 in *B. distachyon* has divergent roles in flowering transition under different day lengths, it would be interesting to know whether *B. distachyon* FTL10 has the same effect on flowering as FT3 in barley or FTL9 in *B. distachyon*. There is lack of *B. distachyon* FTL9 subclade proteins in barley and wheat, but in other grass species outside *Triticeae*, such as rice, sorghum and maize, *B. distachyon* FTL9 has clear orthologs (Supplementary Fig. 12). It has been revealed that *Zea mays CENTRORADIALIS 12 (ZCN12)*, a *FTL9* ortholog in maize, can be induced by a SD treatment, but whether ZCN12 can function as *B. distachyon* FTL9 to compete with the floral activator, ZCN8, to postpone flowering under non-inductive photoperiod is still unknown[40,41]. The *FTL9* ortholog *SbFT8* in sorghum when overexpressed in *A. thaliana* results in an extremely rapid flowering, and can rescue the flowering defects in *A. thaliana ft* mutants under LDs, implicating that FTL9 in *B. distachyon* and *SbFT8* in sorghum may play different roles in activation of flowering[42]. Future work on exploring the biological relevance of FTL9 orthologs in short-day plants may be helpful to explain why *B. distachyon* harbors but wheat and barley lose *FTL9* gene lineage.

As the central flowering pathway hub, FT family protein generally has consistent roles in regulating reproductive development, even though some FT homolog represses this transition[43]. In a previous study, we demonstrated that one FT in temperate grass, FT2, can play opposite roles in floral onset through an alternative splicing strategy, which is simultaneously

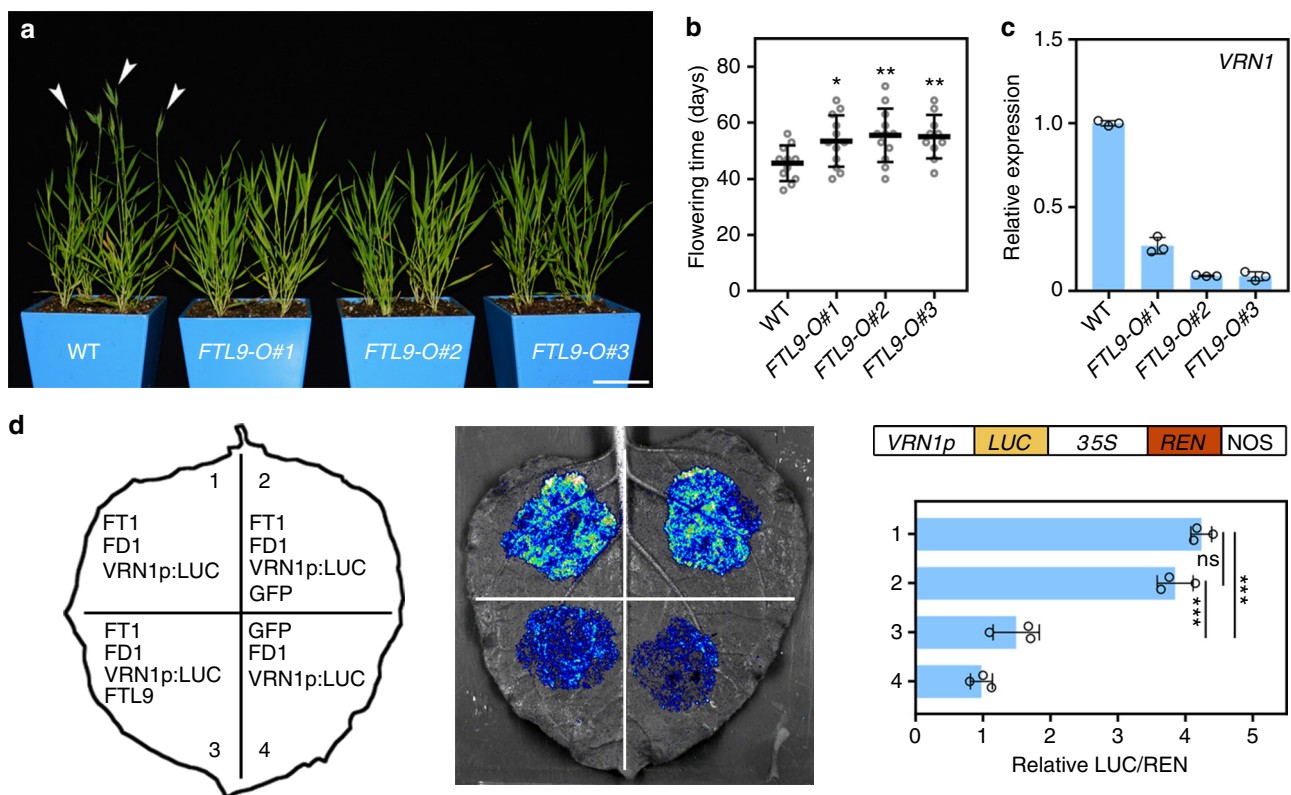

**Fig. 4** Over-expression of *FTL9* results in a flowering delay under LDs in *B. distachyon*. **a** Representative phenotypes of three *FTL9* over-expressing lines under LDs. White arrows point to spikes. Scale bar, 5 cm. **b** Flowering time of wild-type Bd21 and the *FTL9-OE* plants under LDs. At least 10 plants for each line were scored. (Student's *t* test, **$P < 0.01$, *$P < 0.05$). **c** qPCR analysis of flowering downstream gene *VRN1* expression in four-week-old wild-type and the *FTL9-OE* plants under LDs. Error bars indicate s.d. **d** FTL9 attenuates the binding ability of FT1-FAC to *VRN1* cis-elements. Left: Different combinations of effectors and the reporter transiently introduced in *N. benthamiana* leaves. Middle: Representative photograph of firefly luciferase fluorescence signals when the indicated reporter and effectors were introduced in *N. benthamiana* leaves. Right: Relative reporter activity (LUC/REN) in *N. benthamiana* leaves expressing the indicated reporters and effectors. Error bars indicate s.d. (Student's *t* test, ***$P < 0.001$)

generating two isoforms that positively and negatively involved in age flowering pathway, respectively[3]. Together with our present finding that FTL9 has adverse manners in flowering control under different photoperiods, these results suggest that one FT-like protein can be conferred multifaceted roles in heading date in temperate grasses. Because there are more FT close orthologs in temperate grasses than *A. thaliana*, it is interesting to investigate whether other *FT* genes can also perform divergent roles in flowering transition.

In *A. thaliana*, CO is a direct activator of *FT* in the leaf vasculature to trigger flowering under facultative environments[1,37]. However, the CO ortholog in rice, Heading date 1 (Hd1), has dual characters in flowering control, accelerating flowering under SDs through promotion of the Hd3a, the FT ortholog, while suppressing flowering under LDs by repression of Hd3a[44,45]. A recent finding explains the opposite regulatory activity of Hd1 in response to day length that is mediated by DAYS TO HEADING 8, a NUCLEAR FACTOR Y (NF-Y) transcription factor that can physically interact with Hd1[46,47]. Similar as that in rice, a CO ortholog, CO1, in *B. distachyon* can perform dual roles in triggering flowering under LDs, activating *FT1* and inhibiting *FTL9*, although how CO1 can recognize different targets remains unknown. Since NF-Y is able to directly associate with or recruit histone modifiers on *FT* cis-elements[6,48,49], we propose that CO1 in *B. distachyon* may interact with alternative NF-Y transcription factors to recognize different *FT* targets. Further studies are required to explore the specific module by CO1 and NF-Y to

activate *FT1* and repress *FTL9* to trigger flowering under LDs in *B. distachyon*.

Meanwhile, although *FTL9* could be regulated by CO1, the expression of FTL9 is not extremely high in *CO1* knock-down lines under LDs. Given that there are two close CO orthologs in *B. distachyon*, CO2 may function redundantly with CO1 to suppress *FTL9* to affect flowering. Supporting this hypothesis, we did not observe a remarkably precocious flowering in *CO1-RNAi* lines under SDs compared with wild-type plants (Supplementary Fig. 13). Nonetheless, this also may be due to the non-functions of CO1 under SDs in *B. distachyon*, since CO protein has been shown to be degraded by a COP1-mediated ubiquitination complex in the evening in plants[50–52].

It is also possible that FTL9 could be regulated by other modulators in addition to CO-like proteins. VRN2 is such a candidate because VRN2 represses *PPD-H2* has been shown in barley[53,54]. Additionally, *VRN2* is expressed in LD but not in SD, and acts as a repressor of flowering in *B. distachyon*[26,27,55]. Lastly, *VRN2* is also attenuated in the *phyc-1* mutants and our *CO1-RNAi* transgenic lines (Supplementary Fig. 14a, b)[30]. Hence, VRN2 may mediate flowering control via suppression of *FTL9* in *B. distachyon*. Intriguingly, our *CO1* overexpressing *B. distachyon* lines can trigger a premature flowering, even if *VRN2* is prominently enhanced compared with wild-type plants (Supplementary Fig. 14c). In barley, the effects on flowering of CO over-expression depends on VRN2, inducing flowering when VRN2 loses while repressing flowering when VRN2 exists[36]. Thus, CO and VRN2

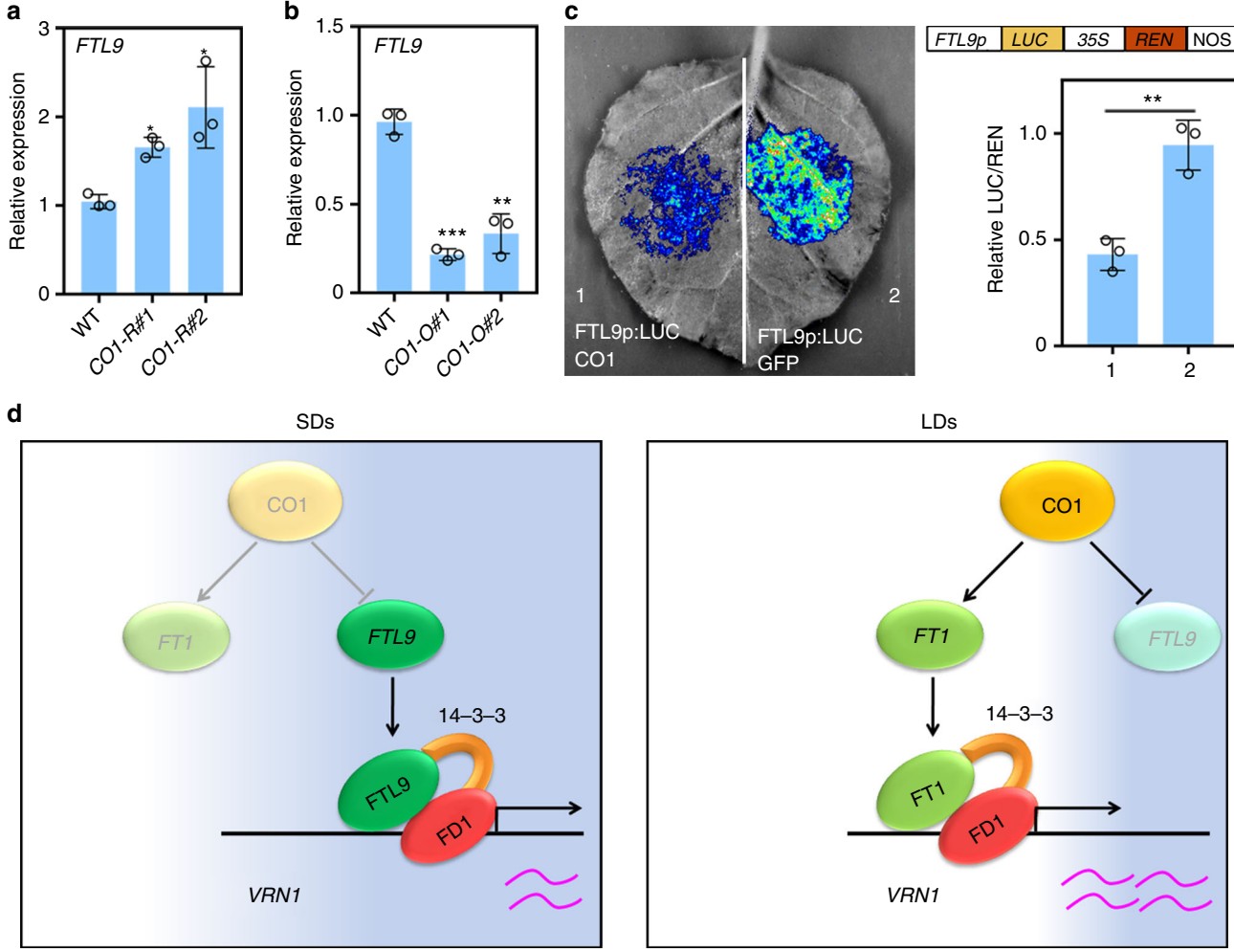

**Fig. 5** CO1 negatively regulates *FTL9* in *B. distachyon*. **a** qPCR analysis of *FTL9* expression in four-week-old wild-type Bd21-3 and two independent *CO1-RNAi* (*CO1-R*) lines under LDs. Error bars indicate s.d. (Student's *t* test, *$P < 0.05$). **b** qPCR analysis of *FTL9* expression in four-week-old wild-type Bd21-3 and two independent *CO1* overexpressing (*CO1-O*) transgenic lines under SDs. Error bars indicate s.d. (Student's *t* test, **$P < 0.01$, ***$P < 0.001$). **c** Examination of CO1 repressive activity to *FTL9* in *N. benthamiana* leaves. Left: Representative photograph of firefly luciferase fluorescence signals when the indicated reporter and effectors were introduced in *N. benthamiana* leaves. Right: Relative reporter activity (LUC/REN) in *N. benthamiana* leaves expressing the indicated reporters and effectors. Error bars indicate s.d. (Student's *t* test, **$P < 0.01$). **d** A working model for the regulatory pathway of flowering initiation in *B. distachyon* under different day lengths. FTL9 can constitute a FAC with FD1 to trigger flowering, but the flowering inductive efficiency of FTL9-FAC is much lower than that of FT1-FAC, thereby resulting in a positive role of FTL9 in plant reproductive transition when *FT1* is not expressed, while a dominant-negative role when FT1 is accumulated. The transcription and stabilization of CO is largely dependent on light in plants. In temperate grass, a CO ortholog, CO1, functions as a activator of *FT1* whereas a repressor of *FTL9*. The decrease of *CO1* mRNA and protein under SDs leads to an inhibition of FT1 and a release of FTL9, which makes plants a delay but an eventual floral onset in the non-inductive photoperiod. When plants grow under LDs, the accumulation of CO1 mRNA and protein is able to lead to a significant increase of *FT1* and a dramatic reduction of *FTL9*, which triggers a rapid flowering

may have different interactions in barely and *B. distachyon*. Further research will be required to seek additional upstream regulators of *FTL9* in the photoperiodic flowering pathway.

## Methods

**Plant growth and flowering time measurements**. *Brachypodium distachyon* plants were sown in soil and grown with the temperature 22 °C/18 °C days/nights and the approximate 12000 Lux light intensities. Flowering time was scored as the number of days from the date of planting in soil to the first day that emergence of the spike was detected. At least 10 plants of the transgenic line and the wild-type plants were used to score flowering time, which was measured in days. Unless stated otherwise, all plants in the study were *B. distachyon* Bd21 accession, grown in growth chambers under LD (16 h of light/8 h of dark) or SD (10 h of light/14 h of dark) conditions. The seedlings or leaves used in the experiments were generally harvested at Zeitgeber time 4.

**Constructs and plant transformation**. To generate *UBI::FTL9*, the full-length coding sequence (CDS) of *FTL9* was amplified from *B. distachyon* Bd21 cDNA

using primers shown in Supplementary Table 1, and cloned into the binary vector pCUBI1390 with a maize ubiquitin promoter. To construct *FTL9* CRISPR-Cas9 system, the sgRNA expression cassette containing *FTL9* target sequence under rice U3 promoter was generated by an overlapping PCR approach. Then, the PCR products and the binary vector pYLCRISPR/Cas9 P$_{Ubi}$-H were ligated in a mixture after simultaneously digested by BsaI (New England Biolabs). The ligated products with sgRNA expression cassettes were directly transformed into *E. coli* DH5α competent cells[56]. Chimeric FTL9:FT1 PEBP was constructed by replacing the residues 23–160 of FTL9 with the residues 25–163 of FT1, and FT1:FTL9 PEBP was constructed by replacing the residues 25–163 of FT1 with the residues 23–160 of FTL9. The full-length CDS of *CO1* was amplified from Bd21 *B. distachyon* cDNA and constructed into pCUBI1390 under a maize ubiquitin promoter to produce CO1 constitutively expressing vector. The *CO1* gene specific region was obtained from *B. distachyon* cDNA, and cloned into the pANDA-β vector to produce a hairpin structure of *CO1* fragment for RNA silencing under maize ubiquitin promoter. All gene overexpression, RNAi and CRISPR/Cas9 clones were validated by sequencing and then transformed into *Agrobacterium tumefaciens* AGL1 strain. *B. distachyon* transformation was based on the infection of embryogenic callus generated from immature embryos with *A. tumefaciens* followed by the selection of transgenic

tissues on media with antibiotic hygromycin[57]. Independent transgenic lines were genotyped using *HPT II*-specific forward and reverse primers.

**Gene expression analysis**. Newly expanded leaves from at least six plants mixed for each indicated line were collected for RNA extraction and gene expression analysis. Total RNA was extracted using RNAiso Plus (Takara) and reversely transcribed using reverse transcriptase (Promega). cDNAs were generated from reverse transcription using Oligo (dT) primers[3]. Real-time PCR was done in triplicates on Step-One Plus real-time PCR system (ABI) with the Power Up SYBR Master Mix (ABI) with three biological replicates. The following cycling conditions were used for Real-time PCR: 2 min at 95 °C, 40 cycles of 10 s at 95 °C, and 40 s at 65 °C, and a final step for melting curve determination (15 s at 95 °C, 1 min at 60 °C, and 15 s at 95 °C). *UBC18* was used as an internal control. Gene expression was calculated using the $2^{\Delta\Delta Ct}$ method. Primers used for gene expression analysis are listed in Supplementary Table 1.

**Phylogenetic analysis**. Phylogenetic trees were generated from the aligned protein sequences using MEGA6. Bootstrap test of phylogeny was performed using the neighbor-joining method from 1000 replications for each branch. Accession numbers for *B. distachyon* genes referred to in this study are as follows; *BdFT1* (Bradi1g48830), *BdFT2* (Bradi2g07070), *BdFTL10* (Bradi2g19670), *BdFT4* (Bradi1g38150), *BdFTL9* (Bradi2g49795), *BdFT6* (Bradi3g08890), *BdFD1* (Bradi4g36587), *BdVRN1* (Bradi3g08340), *BdFUL2* (Bradi1g59250), *BdCO1* (Bradi1g43670), *BdCO2* (Bradi3g56260), *BdVRN2* (Bradi3g10010).

**Yeast two-hybrid assays**. Full-length coding sequences of *FT1*, *FTL9*, *FD1*, and *14-3-3* were amplified from Bd21 *B. distachyon* cDNA by the primers listed in Supplementary Table 1. PCR products were cloned into the *EcoRI* and *BamHI* sites of the DNA-binding domain vector pGBKT7 or the GAL4-activation domain vector, pGADT7. Both vectors were co-transformed into yeast strain AH109 using the lithium acetate-based transformation protocol. The positive colonies were selected SD/-Leu/-Trp (-LT) medium and then used for a growth assay on selective SD/-Leu/-Trp/-His/-Ade medium (-LTHA) supplemented with 1 mM 3-AT. The interactions were observed after 4 days of incubation at 30 °C. The quantitative interaction assays were performed using Chlorophenol red-β-galactopyranoside (Sigma) as substrates[3]. Yeast cells were broken by the freeze/thaw method and mixed with Chlorophenol red-β-D-galactopyranoside in a reaction Buffer. Reactions were stopped by adding 3 mM ZnCl₂, and the color change of the reaction was measured at 578 nm using a UV/Vis spectrophotometer.

**BiFC assays**. Full-length coding sequences of the indicated proteins were constructed into vectors containing either N-terminal or C-terminal-enhanced yellow fluorescence protein fragments. Primers for the constructs are presented in Supplementary Table 1. These resulting constructs were transformed into *A. tumefaciens* strain EHA105, which was then cultured overnight and adjusted to OD600 = 0.8. An equal volume of each culture was mixed together for injection. Sets of combination were co-infiltrated to 4-week-old *N. benthamiana* leaves. YFP fluorescence was observed using a confocal laser scanning microscope (Zeiss, LSM700) 2–3 days later.

**Luciferase imaging assays**. For the firefly LUC complementation imaging assays, the full-length coding sequences of FD1 were ligated with N-terminal fragment of luciferase (nLUC) to form FD1-nLUC. The full-length coding sequences of FT1 and FTL9 were fused with C-terminal fragment of luciferase (cLUC) to form cLUC-FT1 and cLUC-FTL9 to examine the interactions of FT1 and FTL9 with FD1. After a Luciferin (100 μM) spray, the leaves were kept in a dark condition for 5 min before fluorescence observation. The images were taken by a low-light cooled charge-coupled device imaging apparatus (Tanon 5200).

To determine the flowering inductive efficiency of FTL9-FAC and FT1-FAC, equal concentrations and volumes of *A. tumefaciens* strains EHA105 harboring FD1 and the indicated FT or control proteins were mixed with 1/3 amount of *A. tumefaciens* containing 3 kb *VRN1* promoter-driven LUC constructs, and co-infiltrated into *N. benthamiana* leaves by a syringe. Similarly, equal concentrations and volumes of *A. tumefaciens* having CO1 and control proteins mixed with 1/3 amount of those containing 3 kb *FTL9* promoter-driven LUC constructs were co-introduced into *N. benthamiana* for a detection of the repressive activity of CO1 to FTL9. At least four leaves from independent *N. benthamiana* plants were infiltrated, and one leaf was for fluorescence signal observation and other leaves were punched and put in the −80 °C refrigerator to measure the LUC activity together later. For observation of the fluorescence signal, the injected tobacco leaves were kept in a dark condition for 5 min after a spray of Luciferin (100 μM). The LUC images were taken by the imaging device (Tanon 5200). The LUC and REN activity of infiltrated leaf were measured via the Dual-Luciferase Reporter Assay System (Promega) on a GloMax 96 microplate luminometer (Promega), according to the manufacturer's instructions. Final transcriptional activity was indicated by the ratio of LUC/REN.

**Reporting summary**. Further information on experimental design is available in the Nature Research Reporting Summary linked to this article.

## Data availability

The source data underlying Figs. 1–5 and Supplementary Figs. 1–3, 5–6, 8–11, and 13–14 are provided as a Source Data file. All other data that support the findings of this study are available from the corresponding author upon request.

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

## Acknowledgements

We sincerely thank Dr. Joyce Van Eck for helping us to establish *B. distachyon* transformation system. We also thank Dr. Yaoguang Liu and Richard M. Amasino for the gift of efficient CRISPR/Cas9 gene-editing system and *B. distachyon phyc* mutants, respectively. We appreciate Dr. Xueren Yin for assistance of LUC detection analysis. This work was supported by the National Natural Science Foundation of China (31570226, 91640109, 31700189), Zhejiang Provincial Natural Science Foundation of China (LR16C060001), China Postdoctoral Science Foundation (2017M610364, 2018T110586), the open funds of the State Key Laboratory of Crop Genetics and Germplasm Enhancement (ZW201807) and Hundred-Talent Program of Zhejiang University.

## Author contributions

L.W. conceived and initiated the research. L.W. and Z.Q. designed the experiments. Z.Q., Y.B., S.M. and X.W. performed the experiments. Z.Q., Y.B., J.W., P.D., H.A. and L.W. analyzed the data. L.W. and Z.Q. wrote the article.

## Additional information

**Competing interests:** The authors declare no competing interests.

