## [Peer Review File · Nature Communications]

Reviewers' comments:

Reviewer #1 (Remarks to the Author):

This paper studies the function of a homologue of the flowering time gene FT of Arabidopsis in the model grass Brachypodium. This grass species flowers early in response to long days and late in response to short days. Most FT-like genes in Brachypodium are shown to be upregulated in long days and therefore correlate with rapid flowering, as in Arabidopsis and other grasses. However, here they show that FT3 mRNA exhibits the opposite pattern and is increased highly in short days. They show that this is controlled by genes known to regulate photoperiodic response such as the photoreceptor PhyC, so that in phyC mutants FT3 is expressed highly in long days. Also, the transcription factor CO1, which they show activates FT3 in short days and represses it in long days. Using CRISPR they show that ft3 mutants are late flowering in short days. Also the overexpressor is early flowering in short days and late flowering in long days. They use a transient assay to show that FT3 activates a target effector gene less effectively than FT1 and that it can compete with FT1 in this assay reducing its activity. They produce a model in which in short days CO1 is inactive and FT3 is expressed to promote flowering weakly. Whereas in LDs, CO1 activates FT1 to promote flowering rapidly and represses FT3 to prevent it from competing with FT1.

This paper describes a good genetic and molecular analysis of FT3. It does look convincing that FT3 promotes flowering in short days but weakly. This is therefore similar to RFT in rice, which seems to be a weak promoter of flowering under non-inductive long days in that system. The idea that FT3 is a repressor of flowering under long days in Brachypodium is not really true in wild-type plants as FT3 is not expressed in these conditions and the overexpressor plants are relatively weakly late flowering in long days. I find it also important to look in more detail at the mechanisms of FT3 being a weak promoter of flowering – presumably this is to do with differences in the amino acid sequence that might be predicted by cross species comparisons and tested. I think such experiments would be essential for this journal. Overall, I think this is an interesting set of data, but maybe not novel enough or taken far enough for Nature Communications.

Detailed points:

The title, Abstract and much of the Introduction and text stress the results are relevant for “temperate grasses” but all the data shown are in Brachypodium. So it would be more accurate to emphasise Brachypodium and then discuss the relevance for other species in the Discussion.

They should present a detailed analysis of the FT3 sequence and discuss why it might be a weaker protein than FT1. For this journal, I would also like to see such predictions tested.

I do not like the emphasis on FT3 being a repressor of flowering under long days, including a sub-heading of the Results, because this is only true in artificial overexpressor plants not in wild-type. Although it is fair to say that this does make the repression in long days by CO 1 and PHYC more interesting.

The flowering time of CO1-R lines is shown in Figure S9b and they are late flowering in long days because FT1 is expressed less. But no flowering time is shown in short days. They are expected to be early flowering because FT3 is increased (Figure 4a) but this should be tested.

Reviewer #2 (Remarks to the Author):

The authors characterize a short day (SD) expressed FT-like gene in the temperate grass model Brachypodium distachyon and explore the FTlike gene function in promoting flowering under non-

inductive (SD). Temperate grasses are typically considered long day plants however certain varieties can flower under non-inductive SD. Little is currently known at a molecular level what causes SD flowering in temperate grasses. A QTL analysis between SD flowering and LD flowering barley identified a peak called PPDH2 and FT3 is a candidate gene underlying that QTL. The study of the molecular underpinnings between SD and LD plants is interesting and can lead to improvements in agriculture. The authors do a fairly good job characterizing a SD expressed FT like gene however see below for comments that I hope will improve the current manuscript. There are several places in the manuscript that would greatly benefit by providing more context to the reader of what is currently known about flowering in grasses.

Major comments:

1) Bradi2g49795 which the authors claim is orthologous to FT3 is inaccurate. In fact another gene, Bradi2g19670 (called FT5 in this manuscript) is the ortholog of FT3 (Higgins et al., 2010 ; Helliwell et al., 2016). Bradi2g19670 appears to fall within what looks like the FT3 clade even in the analysis in Supp Fig. 1 however it is difficult to know given the lack of labels in the figure. Please highlight in Supp Fig 1 phylogenetic analysis the FT1, FT2, FT3 and FT5 gene lineages in wheat and barley to aid in interpreting the phylogenetic relationships. It might also be worth bolding Bradi2g49795 in the analysis. Note in the phylogenetic analysis in Supp Fig 1 Bradi2g49795 is orthologous with Os01g54490. This gene in rice is not an ortholog of FT3 in wheat and barley but is referred to as FTL9 (Higgins et al., 2010 ; Helliwell et al., 2016; Faure et al., 2007). Note in Helliwell et al, 2016 Bradi2g19670 falls within the FT3 clade and is located in a syntenic location with FT3 supporting orthology whereas it appears there is not a clear ortholog of Bradi2g49795 in wheat and barley despite having clear orthology with other grass species spanning grass diversification ie rice, maize. This suggests that in Tritaceae there was loss of the FTL9 gene lineage (Bradi2g49795) and a tritaceae specific duplication producing FT5. Note no other grass species outside Tritaceae fall within the FT5 clade also see (Faure et al., 2007).

Perhaps Bradi2g19670 is also expressed in SD? In light of the relationships of the closely related genes Bradi2g49795 and Bradi2g19670 and the fact that Bradi2g19670 is FT3 it might be worth adding the expression of Bradi2g19670 in SD and LD.

2) The model Fig. 4 focuses on SD expressed FT3 turning on VRN1 which enables flowering. However, it can't be ruled out that FT3 might in fact also be involved in turning on FT1 for example. What is the expression of FT1 and FT2 in the FT3 overexpression lines grown in SD similar to what is shown for VRN1 in Fig. 2I? In barley it has been suggested that FT3 may in fact be involved in the induction of FT1 in SD and that FT1 subsequently represses FT3 Kikuchi et al. 2009. In barley cultivars that can flower in SD, HvFT1 has been postulated to be a repressor of HvFT3 because relatively high HvFT3 expression in SD precedes the expression of HvFT1 and, as HvFT1 expression levels increase in SD, HvFT3 expression decreases (Kikuchi et al., 2009). Do you eventually see elevated expression of FT1 in SD? I appreciate that the interpretation of this result is difficult given there is a positive feedback loop between FT1 and VRN1 in *Brachypodium*, but perhaps FT3 is also involved or interferes with the loop and it would be helpful to reader to discuss further to provide more context (Ream et al., 2014)?

3) The pictures of nearly all the plants in this study appear to be stressed or look unhealthy. For example, in Fig 2F half of the plants are yellow and appear spindly. When *Brachypodium* or most grasses are this delayed in flowering results in a very "bushy" phenotype due to increased tillering. In the materials and methods state the light intensity used at plant height which is common in flowering time studies as this can greatly impact the timing of flowering. Also state the type of light (Fluorescence, halogen etc) and the type of fixtures used. Due to the stressed nature of the plants it is difficult to interpret flowering time as stress pathways can greatly influence flowering time. I think it is critical the pictures remain in any future iterations of the manuscript as this allows others to better judge the flowering phenotypes. Was the use of twist ties in the manner

shown in the pictures necessary? It looks like the twist ties greatly reduced the growth of the plants.

4) The explanation of why FT3 might delay flowering in LD needs some clarification as this gene still activates VRN1 although not as strongly as UBIFT1. It seems that UBIFT3 would still activate VRN1 more strongly than in wild type given that at the time of tissue harvest there was no FT1 expression. Please expand in the manuscript to provide clarity.

5) Your work that UBI:FT3 delays flowering under inductive photoperiods is the opposite of what others have reported in rice and barley (Kikuchi et al 2009; Mulki et al., 2018). Specifically overexpression of HvFT3 in rice accelerated flowering (Kikuchi et al 2009) and overexpression of HvFT3 in barley resulted in more rapid flowering under inductive LD but not under non-inductive SD (Mulki et al., 2018). Please place your work in context with what has already been done in the discussion section of the manuscript. Why might there be a difference in *Brachypodium*? Granted the other studies are of FT3 which is not the ortholog of the gene you overexpressed. So it could be there are different functions between FT3 and FTL9 despite being closely related paralogs? (see 1 in this review).

6) For the CO work only 2 independent lines were used (Fig. 4), it is common to include 3 independent transgenic lines which allows a better assessment of the phenotype. Were any additional transgenic lines generated but not included for some reason in the manuscript?

Minor Comments:

7) It might be worth discussing which FT-like genes provide florigen activity in SD plants like sorghum or maize? FT3 is orthologous with CN8 from sorghum and maize which along with CN12 are the florigens in those species (Meng et al., 2011 and Lazakis et al., 2011)

8) The authors analyze expression of Bradi2g49795 and FT1 in a phyC mutant however there is no mention of where the phyC mutant came from, I assume Woods et al., 2014 but it would be worth explicitly stating so in the figure legend, main text and/or the materials and methods. Also in Woods et al., 2014 three alleles were isolated, which allele was used in this study?

9) In addition to mentioning that FT1 is greatly attenuated in the phyC mutant also mention during the CO1 section of the paper that CO1 expression is also greatly attenuated in the phyC background (Woods et al., 2014). Which is consistent with the connection you have made that CO1 is a repressor FT3 in LD. Specifically, low CO1 levels in the phyC mutant contributes to the elevated expression of FT3.

10) Line 261-262 I am confused by "...the expression of FT3 is not diminished in CO1 knock down lines" and there is subsequent discussion that this could be due to redundancy with CO2. However, you have shown convincingly that CO1 represses FT3 in LD by showing FT3 expression is elevated in the CO1 knockdown lines in LD and reduced in the CO1 overexpression lines in SD. Thus one would not expect FT3 to be diminished in the CO1 knock down lines.

11) Line 264-265 You bring up the excellent point that other genes expressed in LD might also regulate FT3. Please discuss further that VRN2 is an excellent candidate gene that might repress FT3 in *Brachypodium* given this has already been shown in barley (Casao et al., 2011a; 2011b). Additionally, VRN2 is expressed in LD but not in SD in *Brachypodium* and acts as a repressor of flowering (Casao et al., 2011a; 2011b; Woods et al., 2016). Lastly VRN2 is also attenuated in the phyC mutant (Woods et al., 2014).

This connects with your CO1 work as CO1 has been suggested as a regulator of VRN2 in LD in barley and wheat (eg Mulki et al., 2016, already cited in the manuscript).

12) What is the expression of VRN2 in the CO1 knock down and overexpression lines? This is important given VRN2 has been shown to repress FT3 in barley and thus you can establish the

CO1-VRN2-FT3 connection in Brachypodium. Furthermore, VRN2 has been shown to be a repressor of FT1 in brachypodium Woods et al., 2016 so it is reasonable to think it might also repress other FT genes and might be worth noting in the manuscript. Furthermore, VRN2 and FT are also candidate genes underlying differences in flowering time between rapid and delayed flowering accessions in Brachypodium (Woods et al., 2017; Bettgenhaeuser et al., 2017) which might be worth pointing out in the introduction or discussion. Highlights conserved aspect that genes important in flowering in wheat/barley are also important to flowering in Brachypodium.

13) Please spell out what L1S, L3S, L7S, L14S mean in the figure legend for Fig1C

14) In Fig1a state the developmental stage of the plants when tissue was harvested in LD and SD. Is FT1 or FT2 expression elevated in SD grown plants when they are close to flowering in SD? Please state the developmental stage in the figure legends for all expression analyses. The developmental stage plays a huge role in gene expression and is critical in the replication of studies.

15) Line 277-278 the authors state leaves used in experiments were harvested at zt4. This is close to the middle of the photoperiod for SD grown plants but not for LD grown plants. Given that many genes peak in the middle of the photoperiod was zt4 also used for LD studies?

16) Line 280 and 289 state the Brachypodium accession used to clone FT3 and CO1. Recently it has been shown that there is striking sequence variation in the Brachypodium pan-genome (Gordon et al., 2017).

17) Line 153-154 not only does FT1 flower in regeneration media, overexpression of FT1 in Brachypodium results in extreme rapid flowering ~ 20 days even in 10h or 8h SD when wild type take >130 days to flower (Woods et al., 2017) and this should also be mentioned here as this is a striking contrast to the slightly florigenic activity of FT3 in SD.

18) I would suggest calling UBI:FT3 a weak repressor of flowering under LD instead of simply calling it a repressor as this description better matches the phenotype. Adjust figure 3 legend for example.

References:

Casao MC, Karsai I, Igartua E, Gracia MP, Veisz O, Casas AM (2011a) Adaptation of barley to mild winters: A role for PPDH2. *BMC Plant Biol* 11: 164

Casao MC, Igartua E, Karsai I, Lasa JM, Gracia MP, Casas AM (2011b) Expression analysis of vernalization and day-length response genes in barley (*Hordeum vulgare* L.) indicates that VRNH2 is a repressor of PPDH2 (HvFT3) under long days. *Journal of Experimental Botany* 62: 1939–1949

Mulki M, Bi X, von Korff M (2018) FLOWERING LOCUS T3 controls spikelet initiation but not floral development. *Plant Physiology* DOI: <https://doi.org/10.1104/pp.18.00236>

Faure S, Higgins J, Turner A, Laurie DA (2007) The FLOWERING LOCUS T-Like Gene Family in Barley (*Hordeum vulgare*). *Genetics* 176: 599–609

Lazakis CM, Coneva V, Colasanti J (2011) ZCN8 encodes a potential orthologue of Arabidopsis FT florigen that integrates both endogenous and photoperiod flowering signals in maize. *Journal of Experimental Botany* 62: 4833–4842

Meng X, Muszynski MG, Danilevskaya ON (2011) The FT-Like ZCN8 Gene Functions as a Floral Activator and Is Involved in Photoperiod Sensitivity in Maize. *THE PLANT CELL ONLINE* 23: 942–960

Woods DP, McKeown MA, Dong Y, Preston JC, Amasino RM (2016) Evolution of VRN2/Ghd7-Like Genes in Vernalization-Mediated Repression of Grass Flowering. *PLANT PHYSIOLOGY* 170: 2124–2135

Woods DP, Ream TS, Bouché F, Lee J, Thrower N, Wilkerson C, Amasino RM (2017) Establishment of a vernalization requirement in *Brachypodium distachyon* requires REPRESSOR OF VERNALIZATION1. *Proc Natl Acad Sci USA* 114: 201700536–15

Bettgenhaeuser J, Corke FMK, Opanowicz M, Green P, Hernández-Pinzón I, Doonan JH, Moscou MJ (2017) Natural variation in *Brachypodium* links vernalization and flowering time loci as major flowering determinants. *Plant Physiol* 173: 256–268

Woods DP, Bednarek R, Bouché F, Gordon SP, Vogel JP, Garvin DF, Amasino RM (2017) Genetic architecture of flowering-time variation in *Brachypodium distachyon*. *Plant Physiol* 173: 269–279

Responses to Reviewers' Comments

We wish to express our deep appreciation for the reviewers' constructive comments to improve our manuscript. We have performed additional experiments, added some explanations, and accordingly revised the manuscript. We hope that we have addressed all the concerns made by the reviewers. The point-by-point responses to the reviewers' comments are listed in detail below:

Reviewer #1

This reviewer commented that "This paper describes a good genetic and molecular analysis of FT3". The major concern of this reviewer is the reason that why FT5 (previously termed FT3, see below) is a weaker protein than FT1. We compared and tested PEBP domain of FT1 and FT5, and found that the differences of PEBP in FT1 and FT5 is critical to their individual flowering initiation activity. We addressed the reviewer's comments in detail below.

(From the referee) This paper studies the function of a homologue of the flowering time gene FT of Arabidopsis in the model grass Brachypodium. This grass species flowers early in response to long days and late in response to short days. Most FT-like genes in Brachypodium are shown to be upregulated in long days and therefore correlate with rapid flowering, as in Arabidopsis and other grasses. However, here they show that FT3 mRNA exhibits the opposite pattern and is increased highly in short days. They show that this is controlled by genes known to regulate photoperiodic response such as the photoreceptor PhyC, so that in phyC mutants FT3 is expressed highly in long days. Also, the transcription factor CO1, which they show activates FT3 in short days and represses it in long days. Using CRISPR they show that ft3 mutants are late flowering in short days. Also the overexpressor is early flowering in short days and late flowering in long days. They use a transient assay to show that FT3 activates a target effector gene less effectively than FT1 and that it can compete with FT1 in this assay reducing its activity. They produce a model in which in short days CO1 is inactive and FT3 is expressed to promote flowering weakly. Whereas in LDs, CO1 activates FT1 to promote flowering rapidly and represses FT3 to prevent it from competing with FT1.

This paper describes a good genetic and molecular analysis of FT3. It does look convincing that FT3 promotes flowering in short days but weakly. This is therefore similar to RFT in rice, which seems to be a weak promoter of flowering under non-inductive long days in that system. The idea that FT3 is a repressor of flowering under long days in *Brachypodium* is not really true in wild-type plants as FT3 is not expressed in these conditions and the overexpressor plants are relatively weakly late flowering in long days. I find it also important to look in more detail at the mechanisms of FT3 being a weak promoter of flowering – presumably this is to do with differences in the amino acid sequence that might be predicted by cross species comparisons and tested. I think such experiments would be essential for this journal. Overall, I think this is an interesting set of data, but maybe not novel enough or taken far enough for Nature Communications.

Detailed points:

The title, Abstract and much of the Introduction and text stress the results are relevant for "temperate grasses" but all the data shown are in *Brachypodium*. So it would be more accurate to emphasise *Brachypodium* and then discuss the relevance for other species in the Discussion.

Response: Thank you for pointing this out. Yes, we agree that almost all data provided by us are relevant to *Brachypodium*, thus we emphasized more on *Brachypodium* rather than temperate grasses in the revised manuscript. As the reviewer suggested, we changed "temperate grasses" to "*Brachypodium*" in title and some text in the revised manuscript to be more accurate.

(From the referee) They should present a detailed analysis of the FT3 sequence and discuss why it might be a weaker protein than FT1. For this journal, I would also like to see such predictions tested.

Response: Firstly, we artificially swapped FT5 (named FT3 in our previous manuscript, see our response to reviewer 2 below) and FT1 PEBP domain and transiently introduced into *Nicotiana benthamiana* leaf cells to compare the flowering initiation ability between the original FT5 and the altered FT5. We found that the VRN1p-LUC activation by FT5 could be significantly enhanced when the FT5 PEBP domain was changed to FT1 PEBP domain, whereas VRN1p-LUC activation by FT1 could be significantly compromised when the FT1 PEBP domain was changed to FT5 PEBP domain, suggesting that the

differences of PEBP domain of FT5 and FT1 is critical for their divergent flowering initiation activity (Supplemental Figure 7a and 7b).

Subsequently, We compared the amino acids of FT5 with other FTs that are known with a high flowering initiation activity in long-day plant *Brachypodium* and *Arabidopsis*, including BdFT1, BdFT2, AtFT and AtTSF. We assumed that the specific amino acids of FT5 at PEBP domain that are completely different with those in other FTs might be responsible for FT5 mild flowering activity (Supplemental Figure 7c). However, when we swapped Lysine (Lys, K) at residue 128 in FT5, the sole amino acid that is entirely distinct from that in other FTs at PEBP domain, to Arginine (Arg R), we didn't observe a conspicuous alteration of VRN1p-LUC signal compared with the original FT5 (Supplemental Figure 7d), suggesting that only one different amino acid at residue 128 cannot confer the specific flowering initiation capacity of FT5, and whether a large fragment or other amino acids in FT5 are the cause of its particular flowering action requires further determined. As test and confirmation of so many amino acids is huge work, time consuming and beyond scope of this manuscript, it should be another independent story (Hanzawa et al., 2005; Ahn et al., 2006; Ho and Weigel, 2014), and we would like to do this in the future. Please see our additional results in Supplemental Figure 7 in the revised manuscript.

(From the referee) I do not like the emphasis on FT3 being a repressor of flowering under long days, including a sub-heading of the Results, because this is only true in artificial overexpressor plants not in wild-type. Although it is fair to say that this does make the repression in long days by CO 1 and PHYC more interesting.

Response: Yes, we should not emphasis on FT5 being a repressor of flowering under long days only based on the observation of flowering performance of FT5 over-expression plants. We have changed the subtitle to "FT5 is likely to be a negative regulator of flowering under LDs" and the figure legend 3 title to " Overexpression of FT5 results in a flowering delay under LDs

in *B. distachyon* " in the revised manuscript.

(From the referee) The flowering time of CO1-R lines is shown in Figure S9b and they are late flowering in long days because FT1 is expressed less. But no flowering time is shown in short days. They are expected to be early flowering because FT3 is increased (Figure 4a) but this should be tested.

Response: Actually, we determined the flowering time in CO1 RNAi lines under SDs, but we did not observe a profoundly precocious flowering in *CO1-RNAi* lines compared with wild-type plants. Because CO protein is known to be not stable under SDs and could be degraded by a COP1-mediated ubiquitination complex in the evening (Jang et al., 2008; Liu et al., 2008; Zuo et al., 2011), CO may not play a role in SDs. Therefore, it is reasonable that there is a similar flowering performance of wild-type and *CO1-RNAi* lines under SDs.

Reviewer #2

This reviewer commented that "The study of the molecular underpinnings between SD and LD plants is interesting and can lead to improvements in agriculture. The authors do a fairly good job characterizing a SD expressed FT like gene". We addressed this reviewer's points in detail below.

(From the referee) The authors characterize a short day (SD) expressed FT-like gene in the temperate grass model *Brachypodium distachyon* and explore the FTlike gene function in promoting flowering under non-inductive (SD). Temperate grasses are typically considered long day plants however certain varieties can flower under non-inductive SD. Little is currently known at a molecular level what causes SD flowering in temperate grasses. A QTL analysis between SD flowering and LD flowering barley identified a peak called PPDH2 and FT3 is a candidate gene underlying that QTL. The study of the molecular underpinnings between SD and LD plants is interesting and can lead to improvements in agriculture. The authors do a fairly good job characterizing a SD expressed FT like gene however see below for comments that I hope will improve the current manuscript. There are several places in the

manuscript that would greatly benefit by providing more context to the reader of what is currently known about flowering in grasses.

Major comments:

1) Bradi2g49795 which the authors claim is orthologous to FT3 is inaccurate. In fact another gene, Bradi2g19670 (called FT5 in this manuscript) is the ortholog of FT3 (Higgins et al., 2010 ; Helliwell et al., 2016). Bradi2g19670 appears to fall within what looks like the FT3 clade even in the analysis in Supp Fig. 1 however it is difficult to know given the lack of labels in the figure. Please highlight in Supp Fig 1 phylogenetic analysis the FT1, FT2, FT3 and FT5 gene lineages in wheat and barley to aid in interpreting the phylogenetic relationships. It might also be worth bolding Bradi2g49795 in the analysis. Note in the phylogenetic analysis in Supp Fig 1 Bradi2g49795 is orthologous with Os01g54490. This gene in rice is not an ortholog of FT3 in wheat and barley but is referred to as FTL9 (Higgins et al., 2010 ; Helliwell et al., 2016; Faure et al., 2007). Note in Helliwell et al, 2016 Bradi2g19670 falls within the FT3 clade and is located in a syntenic location with FT3 supporting orthology whereas it appears there is not a clear ortholog of Bradi2g49795 in wheat and barley despite having clear orthology with other grass species spanning grass diversification ie rice, maize. This suggests that in Tritaceae there was loss of the FTL9 gene lineage (Bradi2g49795) and a tritaceae specific duplication producing FT5. Note no other grass species outside Tritaceae fall within the FT5 clade also see (Faure et al., 2007).

Perhaps Bradi2g19670 is also expressed in SD? In light of the relationships of the closely related genes Bradi2g49795 and Bradi2g19670 and the fact that Bradi2g19670 is FT3 it might be worth adding the expression of Bradi2g19670 in SD and LD.

Response: Yes, we agree the review that Bradi2g19670 is the direct ortholog of FT3 in wheat and barley based on the protein similarity and syntenic location. We termed Bradi2g49795 as FT3 previously based on a literature before (Lv et al., 2014). After further careful phylogenetic analysis of Bradi2g19670, Bradi2g49795 and other barley and wheat FT3-like genes (Supplemental Figure 1a), we changed the name of Bradi2g19670 to FT3 and Bradi2g49795 to FT5, respectively, as the reviewer suggested in the revised manuscript (Faure et al., 2007; Higgins et al., 2010; Halliwell et al., 2016). We

marked all FTs clearly in our revised figures, clearly stated and discussed the relationship of FT3 and FT5 in the revised text.

This reviewer asked us to check the expression pattern of Bradi2g19670 (FT3) in *Brachypodium* under SDs and LDs. Actually, we had examined the expressions of FT3 in the previous manuscript (Figure 1 in the manuscript), and found that FT3 have a similar expression pattern with FT1 at ZT4, highly expressed under LDs and lowly expressed under SDs (Figure 1 in the manuscript), although FT3 is higher under SDs than LDs at some other Zeitgeber time (unpublished data). *FT5* is the only *FT-like* gene that is much higher at any time of the day under SDs than LDs in *Brachypodium* (Figure 1 and Supplemental Figure 2 in the manuscript).

(From the referee) 2) The model Fig. 4 focuses on SD expressed FT3 turning on VRN1 which enables flowering. However, it can't be ruled out that FT3 might in fact also be involved in turning on FT1 for example. What is the expression of FT1 and FT2 in the FT3 overexpression lines grown in SD similar to what is shown for VRN1 in Fig. 2I? In barley it has been suggested that FT3 may in fact be involved in the induction of FT1 in SD and that FT1 subsequently represses FT3 Kikuchi et al. 2009. In barley cultivars that can flower in SD, HvFT1 has been postulated to be a repressor of HvFT3 because relatively high HvFT3 expression in SD precedes the expression of HvFT1 and, as HvFT1 expression levels increase in SD, HvFT3 expression decreases (Kikuchi et al., 2009). Do you eventually see elevated expression of FT1 in SD? I appreciate that the interpretation of this result is difficult given there is a positive feedback loop between FT1 and VRN1 in *Brachypodium*, but perhaps FT3 is also involved or interferes with the loop and it would be helpful to reader to discuss further to provide more context (Ream et al., 2014)?

Response: We checked the expressions of FT1 under SDs in *Brachypodium*, and found that FT1 could not be accumulated to a high level eventually (Figure 1 below), different from FT5 which is expressed much high under SDs. We also determined the expressions of FT1 in FT5 overexpressing lines under SDs as the reviewer suggested, and found a similar expression of FT1 in

FT5-OE and wild-type plants under SDs. These results indicated that the induction of flowering under SDs by FT5 is the result of an increase of VRN1 and FUL2, neither by a direct trigger nor an indirect trigger of FT1 (Supplemental Figure 6).

Figure 1 *FT1* expression along with plant development under SDs in *B. distachyon*

(From the referee) 3) The pictures of nearly all the plants in this study appear to be stressed or look unhealthy. For example, in Fig 2F half of the plants are yellow and appear spindly. When Brachypodium or most grasses are this delayed in flowering results in a very "bushy" phenotype due to increased tillering. In the materials and methods state the light intensity used at plant height which is common in flowering time studies as this can greatly impact the timing of flowering. Also state the type of light (Fluorescence, halogen etc) and the type of fixtures used. Due to the stressed nature of the plants it is difficult to interpret flowering time as stress pathways can greatly influence flowering time. I think it is critical the pictures remain in any future iterations of the manuscript as this allows others to better judge the flowering phenotypes. Was the use of twist ties in the manner shown in the pictures necessary? It looks like the twist ties greatly reduced the growth of the plants.

Response: Brachypodium is a long-day plant, and can flower approximate 5 month after planting in soil under our 10h/14h short-day conditions. Since the old short-day Brachypodium was too high and too big to keep standing, we used twist ties in the very late stage under SDs to take a photograph, so the

growth of the plants are not affected by the twist ties. As the reviewer suggested, we provided our light information in detail in the revised methods section.

(From the referee) 4) The explanation of why FT3 might delay flowering in LD needs some clarification as this gene still activates *VRN1* although not as strongly as *UBIFT1*. It seems that *UBIFT3* would still activate *VRN1* more strongly than in wild type given that at the time of tissue harvest there was no *FT1* expression. Please expand in the manuscript to provide clarity.

Response: Our previous finding showed that *FT1* could be expressed when plants are 2-weeks-old in LDs, although the expression level is lower than that of when the plants grow older (Wu et al., 2013). Since the flowering inductive activity of *FT3* is much lower than *FT1*, *UBIFT3* can still repress *VRN1* in the early stage under LDs.

(From the referee) 5) Your work that *UBI:FT3* delays flowering under inductive photoperiods is the opposite of what others have reported in rice and barley (Kikuchi et al 2009; Mulki et al., 2018). Specifically overexpression of *HvFT3* in rice accelerated flowering (Kikuchi et al 2009) and overexpression of *HvFT3* in barley resulted in more rapid flowering under inductive LD but not under non-inductive SD (Mulki et al., 2018). Please place your work in context with what has already been done in the discussion section of the manuscript. Why might there be a difference in *Brachypodium*? Granted the other studies are of *FT3* which is not the ortholog of the gene you overexpressed. So it could be there are different functions between *FT3* and *FTL9* despite being closely related paralogs? (see 1 in this review).

Response: As the review stated above, *Bradi2g19670* (*FT3*) should be the real ortholog of *HvFT3* in *Brachypodium*. In this study, we performed the research on *Bradi2g49795* (*FT5*), which places in an independent subgroup in the phylogenetic tree (Supplemental Figure 1a). Therefore, *FT5* may have different roles in flowering control in *Brachypodium*, and we experimentally demonstrated this. We also discussed the relationship between *FT3* and *FT5*

in other grasses in the revised manuscript.

(From the referee) 6) For the CO work only 2 independent lines were used (Fig. 4), it is common to include 3 independent transgenic lines which allows a better assessment of the phenotype. Were any additional transgenic lines generated but not included for some reason in the manuscript?

Response: Since transformation of *Brachypodium* is much more complicated than that of *Arabidopsis*, and pretty time-consuming, we only obtained eight CO1-OE plants. CO1 is significantly enhanced among six of them. For CO1-RNAi constructs, we obtained six transgenic lines and detected dramatic reductions of CO1 in three of them compared with wild-type plants (Supplemental Figure 9b and 9c). We randomly chose two independent CO1-OE and CO1-RNAi lines for further studies due to the space limitations of our growth chamber. You can see the two independent CO1-OE and CO1-RNAi lines shown in Figure 4 have similar performance in flowering onset, thus the assessment of the phenotype is reliable (Supplemental Figure 10 and 11).

Minor Comments: (From the referee) 7) It might be worth discussing which FT-like genes provide florigen activity in SD plants like sorghum or maize? FT3 is orthologous with CN8 from sorghum and maize which along with CN12 are the florigens in those species (Meng et al., 2011 and Lazakis et al., 2011)

Response: We made a phylogenetic tree of FTs including sorghum or maize (Supplemental Figure 12), and added some discussions of their FT5 orthologs in the revised manuscript.

(From the referee) 8) The authors analyze expression of Bradi2g49795 and FT1 in a phyC mutant however there is no mention of where the phyC mutant came from, I assume Woods et al., 2014 but it would be worth explicitly stating so in the figure legend, main text and/or the materials and methods. Also in Woods et al., 2014 three alleles were isolated, which allele was used in this study?

Response: Yes, we obtained *Phyc* mutants from Amasino lab (Woods et al., 2014). We used homozygous *phyc-1* to examine the indicated gene expressions. In the revised manuscript, we added the recourses of *phyc-1* we used in the research and acknowledged Dr. Amasino for a gift of *phyc* mutant seeds. We are sorry to forget to mention it before.

(From the referee) 9) In addition to mentioning that FT1 is greatly attenuated in the *phyC* mutant also mention during the CO1 section of the paper that CO1 expression is also greatly attenuated in the *phyC* background (Woods et al., 2014). Which is consistent with the connection you have made that CO1 is a repressor FT3 in LD. Specifically, low CO1 levels in the *phyC* mutant contributes to the elevated expression of FT3.

Response: Yes, we detected a profound decrease of CO1 in *phyc-1* mutants (Supplemental Figure 3b). Therefore, as mentioned by this reviewer, it is possible that the elevated expression of FT5 in *phyc* may be result of the low expression of CO1.

(From the referee) 10) Line 261-262 I am confused by "...the expression of FT3 is not diminished in CO1 knock down lines" and there is subsequent discussion that this could be due to redundancy with CO2. However, you have shown convincingly that CO1 represses FT3 in LD by showing FT3 expression is elevated in the CO1 knockdown lines in LD and reduced in the CO1 overexpression lines in SD. Thus one would not expect FT3 to be diminished in the CO1 knock down lines.

Response: Sorry for the mistake. We changed "the expression of FT3 is not diminished in CO1 knock-down lines" to "the expression of FT5 is not extremely high in CO1 knock-down lines".

(From the referee) 11) Line 264-265 You bring up the excellent point that other genes expressed in LD might also regulate FT3. Please discuss further that VRN2 is an excellent candidate gene that might repress FT3 in Brachypodium given this has already been shown in barley (Casao et al., 2011a; 2011b). Additionally, VRN2 is expressed in LD but not in SD in Brachypodium and acts as a repressor of flowering (Casao et al., 2011a; 2011b; Woods et al.,

2016). Lastly VRN2 is also attenuated in the phyC mutant (Woods et al., 2014). This connects with your CO1 work as CO1 has been suggested as a regulator of VRN2 in LD in barley and wheat (eg Mulki et al., 2016, already cited in the manuscript).

Response: We added some discussions regarding regulation of FT5 by VRN2 as the reviewer suggested.

(From the referee) 12) What is the expression of VRN2 in the CO1 knock down and overexpression lines? This is important given VRN2 has been shown to repress FT3 in barley and thus you can establish the CO1-VRN2-FT3 connection in Brachypodium. Furthermore, VRN2 has been shown to be a repressor of FT1 in brachypodium Woods et al., 2016 so it is reasonable to think it might also repress other FT genes and might be worth noting in the manuscript. Furthermore, VRN2 and FT are also candidate genes underlying differences in flowering time between rapid and delayed flowering accessions in Brachypodium (Woods et al., 2017; Bettgenhaeuser et al., 2017) which might be worth pointing out in the introduction or discussion. Highlights conserved aspect that genes important in flowering in wheat/barley are also important to flowering in Brachypodium.

Response: We detected *VRN2* expressions in *CO1* knock-down and overexpression lines, and found that *VRN2* is decreased in *CO1*-RNAi lines whereas dramatically increased in *CO1*-OE lines (Supplemental Figure 13). We added some discussions of *VRN2* and its possible regulation of FT5 as the reviewer suggested.

(From the referee) 13) Please spell out what L1S, L3S, L7S, L14S mean in the figure legend for Fig1C

Response: We added this in the revised figure legend of Fig1c.

(From the referee) 14) In Fig1a state the developmental stage of the plants when tissue was harvested in LD and SD. Is FT1 or FT2 expression elevated in SD grown plants when they are close to flowering in SD? Please state the developmental stage in the figure legends for all expression analyses. The developmental stage plays a huge role in gene expression and is critical in the replication of studies.

Response: FT1 and FT2 is not obviously elevated in SD grown plants when they are close to flowering in SD (Figure 1 above)(Wu et al., 2013). We added

the introduction of developmental stage of plant material in each revised figure legend.

(From the referee) 15) Line 277-278 the authors state leaves used in experiments were harvested at zt4. This is close to the middle of the photoperiod for SD grown plants but not for LD grown plants. Given that many genes peak in the middle of the photoperiod was zt4 also used for LD studies?

Response: We collected plant leaves at ZT4 for experiments both under SDs and LDs for comparing gene expressions at the same time of the day.

(From the referee) 16) Line 280 and 289 state the *Brachypodium* accession used to clone FT3 and CO1. Recently it has been shown that there is striking sequence variation in the *Brachypodium* pan-genome (Gordon et al., 2017).

Response: Thank you for pointing this out. We used Bd21 accession to clone *FT5* and *CO1* genes. We indicated this in the revised method section.

(From the referee) 17) Line 153-154 not only does FT1 flower in regeneration media, overexpression of FT1 in *Brachypodium* results in extreme rapid flowering ~ 20 days even in 10h or 8h SD when wild type take >130 days to flower (Woods et al., 2017) and this should also be mentioned here as this is a striking contrast to the slightly florigenic activity of FT3 in SD.

Response: We added this in the revised manuscript and also referred the literature.

(From the referee) 18) I would suggest calling UBI:FT3 a weak repressor of flowering under LD instead of simply calling it a repressor as this description better matches the phenotype. Adjust figure 3 legend for example.

Response: We changed the figure 3 legend as the reviewer suggested.

Reference:

Ahn, J.H., Miller, D., Winter, V.J., Banfield, M.J., Lee, J.H., Yoo, S.Y., Henz, S.R., Brady, R.L., and Weigel, D. (2006). A divergent external loop confers antagonistic activity on floral regulators FT and TFL1. *EMBO J* 25, 605-614.

- Faure, S., Higgins, J., Turner, A., and Laurie, D.A. (2007). The FLOWERING LOCUS T-like gene family in barley (*Hordeum vulgare*). *Genetics* 176, 599-609.
- Halliwell, J., Borrill, P., Gordon, A., Kowalczyk, R., Pagano, M.L., Saccomanno, B., Bentley, A.R., Uauy, C., and Cockram, J. (2016). Systematic Investigation of FLOWERING LOCUS T-Like Poaceae Gene Families Identifies the Short-Day Expressed Flowering Pathway Gene, TaFT3 in Wheat (*Triticum aestivum* L.). *Front Plant Sci* 7, 857.
- Hanzawa, Y., Money, T., and Bradley, D. (2005). A single amino acid converts a repressor to an activator of flowering. *Proc Natl Acad Sci U S A* 102, 7748-7753.
- Higgins, J.A., Bailey, P.C., and Laurie, D.A. (2010). Comparative genomics of flowering time pathways using *Brachypodium distachyon* as a model for the temperate grasses. *PLoS ONE* 5, e10065.
- Ho, W.W., and Weigel, D. (2014). Structural features determining flower-promoting activity of *Arabidopsis* FLOWERING LOCUS T. *Plant Cell* 26, 552-564.
- Jang, S., Marchal, V., Panigrahi, K.C.S., Wenkel, S., Soppe, W., Deng, X.W., Valverde, F., and Coupland, G. (2008). *Arabidopsis* COP1 shapes the temporal pattern of CO accumulation conferring a photoperiodic flowering response. *Embo Journal* 27, 1277-1288.
- Liu, L.J., Zhang, Y.C., Li, Q.H., Sang, Y., Mao, J., Lian, H.L., Wang, L., and Yang, H.Q. (2008). COP1-mediated ubiquitination of CONSTANS is implicated in cryptochrome regulation of flowering in *Arabidopsis*. *Plant Cell* 20, 292-306.
- Lv, B., Nitcher, R., Han, X., Wang, S., Ni, F., Li, K., Pearce, S., Wu, J., Dubcovsky, J., and Fu, D. (2014). Characterization of FLOWERING LOCUS T1 (FT1) gene in *Brachypodium* and wheat. *PLoS ONE* 9, e94171.
- Woods, D.P., Ream, T.S., Minevich, G., Hobert, O., and Amasino, R.M. (2014). PHYTOCHROME C Is an Essential Light Receptor for Photoperiodic Flowering in the Temperate Grass, *Brachypodium distachyon*. *Genetics* 198, 397-+.
- Wu, L., Liu, D., Wu, J., Zhang, R., Qin, Z., Li, A., Fu, D., Zhai, W., and Mao, L. (2013). Regulation of FLOWERING LOCUS T by a microRNA in *Brachypodium distachyon*. *Plant Cell* 25, 4363-4377.
- Zuo, Z.C., Liu, H.T., Liu, B., Liu, X.M., and Lin, C.T. (2011). Blue Light-Dependent Interaction of CRY2 with SPA1 Regulates COP1 activity and Floral Initiation in *Arabidopsis*. *Current Biology* 21, 841-847.

Reviewers' comments:

Reviewer #1 (Remarks to the Author):

This is a resubmission of a manuscript I reviewed previously. The authors have seriously considered my comments and have edited the manuscript to take account of them or have included new data where necessary. I comment below on their comments and the changes they made in response to the four points I raised.

They have altered the title, which now reads better. However, it should read "different day-lengths". In other parts of the manuscript the details of the English could be edited during the publication process.

They have included in Supplementary Figure 7 data comparing FT1 and FT5 that improve the manuscript. The comparison of the protein sequences in S7C and the luciferase transient assays in S7A, B and D. Although heterologous assays, these assays help address the functional differences between FT1 and FT5, which I requested. However, these data should be quantified as shown in Figure 3D. At present they are hard to interpret. Also, in Figure S7 and Figure 3D they should say in the legend how often these assays were independently repeated. From what replicates are the error bars in Figure 3D calculated?

They say that they have changed the subtitle "FT5 is likely to be a negative regulator of flowering under LDs." to "Overexpression of FT5 results in a flowering delay under LDs in *B. distachyon*", as requested. However, unfortunately, on Page 5, Line 196 the original subtitle remains. Therefore, this should be changed in the manuscript. Fortunately, the title to Figure 3 is changed as indicated.

They say that "we determined the flowering time in CO1 RNAi lines under SDs, but we did not observe a profoundly precocious flowering in CO1-RNAi lines compared with wild-type plants." And go on to provide an explanation. I suggest that these data are included in the manuscript for the benefit of readers and that the explanation is provided in the Discussion.

Reviewer #2 (Remarks to the Author):

Major Comments:

1) It is unclear why the authors refer to Bradi2g49795, the focal gene in this study, now as BdFT5. As stated in my previous review and is also demonstrated in the phylogenetic analysis in supplemental figure 1, Bradi2g49795 is not orthologous with FT5 or FT3 from wheat and barley. The duplication event producing FT3 and FT5 occurred after Brachypodium split from Tritiaceae. Naming this gene FT5 may confuse readers thinking that this gene is the Brachypodium ortholog of FT5 from wheat and barley which it is not. I suggest using the nomenclature from Higgins et al., 2010 (manuscript focused on Brachypodium and so it is fitting to use and follows the nomenclature established in rice many years ago) which called the gene FTL9.

2) In all figure panels the FT3 name is used even though you had changed the name to FT5 in the text. Adjust the figure panel labeling to reflect the name change! For example, in the Figure 1 legend heading you state that "FT5 is specifically induced under SDs in *B. distachyon*" however in Fig 1a you show that FT3 is the gene induced in SD. All of the figures are labeled with FT3 when they should be FT5 or in future I suggest calling the gene FTL9 see point 1 above.

3) The knockdown of CO1 is an important result as no mutant phenotype have been published in any other temperate grass for CO. As you know there is a closely related CO2 gene in grasses. Please add the expression of CO2 in the amiCO1 line to test the specificity of the CO1 microRNA. It

is fine if the RNAi effects the expression of both genes but is important to know. I think folks in wheat and barley will be very interested in this exciting result! But testing the specificity is key even if they were originally designed in silico to only target CO1.

4) "These results indicated that the induction of flowering under SDs by FT5 is the result of an increase of VRN1 and FUL2, neither by a direct trigger nor an indirect trigger of FT1" from authors response to reviewer.

The earlier flowering in 10h SD cannot be solely attributed to VRN1 or FUL2 because in Brachypodium it has been shown that overexpression of UBI:VRN1 does not result in rapid flowering in 10h or 8h SD (Woods et al., 2017). Furthermore, even in a *ezl1* (catalytic subunit of the polycomb repressive complex) background which results in the elevated expression of both FUL2 and VRN1 in SD still does not result in more rapid flowering (Lomax et al., 2018). Therefore, expression of VRN1 and FUL2 are not sufficient to induce flowering in Brachypodium under SD. This fact would be worth adding to the manuscript and again it is important to include pertinent flowering time information already known in Brachypodium in the manuscript. The above data indicates that the hypothesis that VRN1/FUL2 are what's key for SD flowering is likely not the reason for the more rapid flowering observed in UBIFTL9 or the delay of flowering in the *ftl9* grown in SD. Have you tested if UBI VRN1 or UBIFUL2 is sufficient to induce rapid flowering in SD in your hands? If so, it would be useful to include the data in the manuscript.

5) Given the focus of the paper is about flowering time in Brachypodium it would be worthwhile to add a paragraph in the introduction about what is known about flowering in Brachypodium to provide context/background for the reader. Address to what extent genes involved in flowering time in Brachypodium are conserved with what is known in wheat and barley. This will help readers make connections.

6) State what the developmental stage and what tissue was used (eg third leaf when plants were at the third leaf stage). Thank you for including how old the plants were at time of harvest but perhaps more important is the developmental age for example after 8 weeks of growth in SD or LD what is the developmental stage, eg 8leaves? Was the same leaf harvested or different leaves for expression work. For Fig 1A, this should be stated for all experiments where expression work was done otherwise work can never be properly replicated.

7) Given additional transgenic lines were generated it is important to mention the total of T0 plants received and the T1 plants further analyzed. "We...chose two independent...lines for further study due to the space limitations of our growth chambers". This is a not a sufficient argument to not include a minimum of three independent transgenic events as Brachypodium is quite small and is the standard in the field.

8) "It has been shown that Zea mays CENTRORADIALIS 12 (ZCN12), a FT5 ortholog in maize, can be induced by a SD treatment, but whether ZCN12 can function as B. distachyon FT5 to repress the floral activator, ZCN8, to postpone flowering under non-inductive photoperiod is still unknown (36, 37)." Line 274-275

The above statement is inaccurate the more rapid flowering when ZCN12 is overexpressed occurs even in the presence of ZCN8. More importantly this contrasts with your proposed hypothesis that ZCN12/FT5 becomes a dominant negative in the presence of a more active florigen such as FT1 or CN8. Furthermore, the FT5 ortholog in sorghum when overexpressed in Arabidopsis results in extreme rapid flowering under long day a condition in which FT1 is expressed (Wolabu et al., 2016). Additionally, elevated expression of the FT5 ortholog in sorghum is also able to rescue a *ft1* mutant in Arabidopsis (Wolabu et al., 2016). I think these results need to be mentioned in the discussion alongside your proposed dominant negative model for FT5. Again it is essential that any results are placed into context with other peoples work. There may very well be differences in the function and evolution of FT5 in different grasses and this makes for interesting discussion.

Minor Comments:

8) Line 159 When discussing overexpression of FT1 I recommend as stated in previous review to include Woods et al. 2017 as this is the only study that has shown that overexpression of FT1 results in extremely rapid flowering (20 days) in a number of different photoperiods including 8hrs.

9) It is interesting that overexpression of CO1 results in an acceleration of flowering even though VRN2 expression levels increase. Note in barley when CO1 is overexpressed there is a delay in flowering when there is a functional VRN2 (Mulki et al., 2016). These are important results and should be mentioned in the results section of the manuscript and discussed.

10) Citation 24 and 31 are missing information.

11) Woods et al., 2014 is cited twice as number 25 and 51

12) In future, when responding to reviewers comments please include in your response exactly what was added or changed in the main text. At the very least include the line number in the main text where the change occurs.

Responses to Reviewers' Comments

We would like to sincerely thank the reviewers' for their constructive comments to improve our work. We have added CO₂ expression analysis in *CO1-RNAi* lines and quantified the transient luciferase activation assays in the revised manuscript. We also provide some interpretations and thereby accordingly revised the manuscript as the reviewer suggested. The point-by-point responses to the reviewers' comments are listed in detail below:

Reviewer #1

(From the referee) This is a resubmission of a manuscript I reviewed previously. The authors have seriously considered my comments and have edited the manuscript to take account of them or have included new data where necessary. I comment below on their comments and the changes they made in response to the four points I raised.

They have altered the title, which now reads better. However, it should read "different day-lengths". In other parts of the manuscript the details of the English could be edited during the publication process.

Response: Sorry for the mistake in title. We have corrected to the "different day-lengths", and edited English carefully in the revised manuscript as well.

(From the referee) They have included in Supplementary Figure 7 data comparing FT1 and FT5 that improve the manuscript. The comparison of the protein sequences in S7C and the luciferase transient assays in S7A, B and D. Although heterologous assays, these assays help address the functional differences between FT1 and FT5, which I requested. However, these data should be quantified as shown in Figure 3D. At present they are hard to interpret. Also, in Figure S7 and Figure 3D they should say in the legend how often these assays were independently repeated. From what replicates are the error bars in Figure 3D calculated?

Response: We have quantified the transient luciferase activation assays in the previous Figure S7a, b and d, and integrated them with previous Figure 2n to form a new Figure 3. Please see the revised Figure 3a-d. For transient luciferase activation assays, we independently injected *A.tumefaciens* with the indicated constructs into at least four leaves from different *N. benthamiana*

plants. We chose one leaves to take LUC images, while the other leaves were punched and put in the -80°C refrigerator to measure the LUC activity together later via the Dual-Luciferase Reporter Assay System on a GloMax 96 microplate luminometer. The error bars in luciferase activity quantification calculated based on the results from these independent leaves. We described this in detail in the revised methods section.

(From the referee) They say that they have changed the subtitle "FT5 is likely to be a negative regulator of flowering under LDs." to "Overexpression of FT5 results in a flowering delay under LDs in *B. distachyon*", as requested. However, unfortunately, on Page 5, Line 196 the original subtitle remains. Therefore, this should be changed in the manuscript. Fortunately, the title to Figure 3 is changed as indicated.

Response: We have changed the subtitle to " Over-expression of FT5 leads to a flowering delay under LDs " in the revised manuscript.

(From the referee) They say that "we determined the flowering time in CO1 RNAi lines under SDs, but we did not observe a profoundly precocious flowering in CO1-RNAi lines compared with wild-type plants." And go on to provide an explanation. I suggest that these data are included in the manuscript for the benefit of readers and that the explanation is provided in the Discussion.

Response: We have added these data and explanation in Supplemental Figure 13 and the revised discussion section .

Reviewer #2

(From the referee) Major Comments: 1) It is unclear why the authors refer to Bradi2g49795, the focal gene in this study, now as BdFT5. As stated in my previous review and is also demonstrated in the phylogenetic analysis in supplemental figure 1, Bradi2g49795 is not orthologous with FT5 or FT3 from wheat and barley. The duplication event producing FT3 and FT5 occurred after Brachypodium split from Triticeae. Naming this gene FT5 may confuse readers thinking that this gene is the Brachypodium ortholog of FT5 from wheat and barley which it is not. I suggest using the nomenclature from Higgins et al., 2010 (manuscript focused on Brachypodium and so it is fitting to use and follows the nomenclature

established in rice many years ago) which called the gene FTL9.

Response: Sorry for the confusion. We named Bradi2g49795 FT5 based on three reasons: 1) FT3 (Bradi2g19670) in *B. distachyon* is indeed a direct orthologue of FT3 in wheat and barley from the phylogenetic tree, and we thereby changed Bradi2g49795 to another name in the last revised manuscript. 2) Although FT5 (Bradi2g49795) is not a direct ortholog as that in wheat and barley, we think it would be inconsistent with the previous nomenclature of FTs in temperate grass if we changed it to FTL9 in the text. Actually, most FT orthologs in temperate grass have been termed FT1, FT2, FT3 and so on (Faure et al., 2007; Greenup et al., 2009; Lv et al., 2014; Ream et al., 2014; Halliwell et al., 2016; Qin et al., 2017; Zikhali et al., 2017; Shaw et al., 2018), although there is a work that named FT-like proteins as FTL before (Higgins et al., 2010). You can also see in our figure 1, there is a determination of FT1-FT6 expression, and we feel that if the name FTL9 appears, it seems to be somewhat abrupt and may lead the readers to be confused. 3) It is normal that the gene function can be different, even if the gene name is same in distinct plants.

We asked the editor whether we have to change the name FT5 to FTL9, the editor suggest leaving this up to us. We can readily term it FTL9 as the reviewer suggested, but we think it is better to keep Bradi2g49795 named as FT5 based on the above reasons. To make a clear justification for the name we chose, we clearly described the gene name and gene number at the end of the manuscript. We also introduced and discussed the relationship between this gene and those in other grasses (Higgins et al., 2010) , and referred the literature as the reviewer suggested. Please see the revised introduction and discussion section.

(From the referee) 2) In all figure panels the FT3 name is used even though you had changed the name to FT5 in the text. Adjust the figure panel labeling to reflect the name change! For example, in the Figure 1 legend heading you state that "FT5 is specifically induced under SDs in *B. distachyon*" however in Fig

1a you show that FT3 is the gene induced in SD. All of the figures are labeled with FT3 when they should be FT5 or in future I suggest calling the gene FTL9 see point 1 above.

Response: We are sorry for the mistake in our integration of text and figures in last manuscript. We have now corrected all names in the revised figures.

(From the referee) 3) The knockdown of CO1 is an important result as no mutant phenotype have been published in any other temperate grass for CO. As you know there is a closely related CO2 gene in grasses. Please add the expression of CO2 in the amiCO1 line to test the specificity of the CO1 microRNA. It is fine if the RNAi effects the expression of both genes but is important to know. I think folks in wheat and barley will be very interested in this exciting result! But testing the specificity is key even if they were originally designed in silico to only target CO1.

Response: We have checked the expressions of CO2 in our CO1-RNAi lines and did not find a decrease of CO2 in them compared with wild-type plants. This is expected because we chose the specific region of CO1 to produce double-strand for silencing it. We have added these data in the revised manuscript. Please see the revised Figure S9c.

(From the referee) 4) "These results indicated that the induction of flowering under SDs by FT5 is the result of an increase of VRN1 and FUL2, neither by a direct trigger nor an indirect trigger of FT1" from authors response to reviewer.

The earlier flowering in 10h SD cannot be solely attributed to VRN1 or FUL2 because in Brachypodium it has been shown that overexpression of UBI:VRN1 does not result in rapid flowering in 10h or 8h SD (Woods et al., 2017). Furthermore, even in a ezi1 (catalytic subunit of the polycomb repressive complex) background which results in the elevated expression of both FUL2 and VRN1 in SD still does not result in more rapid flowering (Lomax et al., 2018). Therefore, expression of VRN1 and FUL2 are not sufficient to induce flowering in Brachypodium under SD. This fact would be worth adding to the manuscript and again it is important to include pertinent flowering time information already known in Brachypodium in the manuscript. The above data indicates that the hypothesis that VRN1/FUL2 are what's key for SD flowering is likely not the reason for the more rapid flowering observed in UBIFTL9 or the delay of flowering in the ftl9 grown in SD. Have you tested if UBI VRN1 or UBIFUL2 is sufficient to induce rapid

flowering in SD in your hands? If so, it would be useful to include the data in the manuscript.

Response: We don't have *VRN1* and *FUL2* over-expression plants, so we can't determine the performance of them in SDs. We are now preparing for generating *VRN1* and *FUL2* gain- and loss-of-function plants, thereby may observe their phenotypes under SDs in the future. We agreed with the reviewer that the induction of flowering under SDs by FT5 may not be a direct result of an increase of *VRN1*, and we discussed this in the revised manuscript. Please see the revised discussion section.

(From the referee) 5) Given the focus of the paper is about flowering time in *Brachypodium* it would be worthwhile to add a paragraph in the introduction about what is known about flowering in *Brachypodium* to provide context/background for the reader. Address to what extent genes involved in flowering time in *Brachypodium* are conserved with what is known in wheat and barley. This will help readers make connections.

Response: We have added some introductions of flowering genes in the revised manuscript to provide background for the reader. We are sorry that we cannot describe too much about this because of the space limitations of the introduction section. Please see the revised paragraph 5 in the revised introduction section. We are now thinking about the possibility of writing a review paper to summarize the molecular advances of flowering control recently made in *B. distachyon* and other temperate grasses.

(From the referee) 6) State what the developmental stage and what tissue was used (eg third leaf when plants were at the third leaf stage). Thank you for including how old the plants were at time of harvest but perhaps more important is the developmental age for example after 8 weeks of growth in SD or LD what is the developmental stage, eg 8 leaves? Was the same leaf harvested or different leaves for expression work. For Fig 1A, this should be stated for all experiments where expression work was done otherwise work can never be properly replicated.

Response: We described the developmental stage of plants in the revised manuscript. We harvested newly expanded leaves for gene expression work

and stated this in Figure 1A and Method section as the reviewer suggested.

(From the referee) 7) Given additional transgenic lines were generated it is important to mention the total of T0 plants received and the T1 plants further analyzed. "We...chose two independent...lines for further study due to the space limitations of our growth chambers". This is a not a sufficient argument to not include a minimum of three independent transgenic events as *Brachypodium* is quite small and is the standard in the field.

Response: Sorry, our growth chamber space is indeed very limited, and *Brachypodium* is much bigger than *Arabidopsis*, especially when they grows in SD environments. Because we have to plant many other transgenic lines for additional projects, we generally used two independently confirmed transgenic plants for further studies in depth. Thank you for the reviewer's understanding.

(From the referee) 8) "It has been shown that Zea mays CENTRORADIALIS 12 (ZCN12), a FT5 ortholog in maize, can be induced by a SD treatment, but whether ZCN12 can function as B. distachyon FT5 to repress the floral activator, ZCN8, to postpone flowering under non-inductive photoperiod is still unknown (36, 37)." Line 274-275 The above statement is inaccurate the more rapid flowering when ZCN12 is overexpressed occurs even in the presence of ZCN8. More importantly this contrasts with your proposed hypothesis that ZCN12/FT5 becomes a dominant negative in the presence of a more active florigen such as FT1 or CN8. Furthermore, the FT5 ortholog in sorghum when overexpressed in *Arabidopsis* results in extreme rapid flowering under long day a condition in which FT1 is expressed (Wolabu et al., 2016). Additionally, elevated expression of the FT5 ortholog in sorghum is also able to rescue a ft1 mutant in *Arabidopsis* (Wolabu et al., 2016). I think these results need to be mentioned in the discussion alongside your proposed dominant negative model for FT5. Again it is essential that any results are placed into context with other peoples work. There may very well be differences in the function and evolution of FT5 in different grasses and this makes for interesting discussion.

Response: Sorry, we could not find a reference that described a rapid flowering triggered by an overexpressed ZCN12 in the presence of ZCN8. Thus, it is difficult to determine whether ZCN12 in maize, the ortholog of FT5 in *Brachypodium*, has an opposite roles to FT5 in flowering. Moreover,

although SbFT8, the ortholog of FT5 in sorghum, can result in an extreme rapid flowering under long-day condition when overexpressed in *Arabidopsis*, and can rescue *ft1* mutant in *Arabidopsis*, it is difficult to get a conclusion that SbFT8 can function as a positive regulator of flowering in sorghum as FT in *Arabidopsis*, because this experiment was performed using a heterologous system (Wolabu et al., 2016). In addition, SbFT8 cannot interact with SbFD1 in Y2H and BiFC assays as SbFT1 (Wolabu et al., 2016), suggesting that SbFT8 may have an alternative role in flowering in sorghum. We discussed all these results in the revised manuscript and we think it is normal that the orthologs in different plants may have different functions.

(From the referee) Minor Comments: 8) Line 159 When discussing overexpression of FT1 I recommend as stated in previous review to include Woods et al. 2017 as this is the only study that has shown that overexpression of FT1 results in extremely rapid flowering (20 days) in a number of different photoperiods including 8hrs.

Response: Sorry, we can't find such a reference (Woods et al. 2017) that is the only study that has shown that overexpression of FT1 results in extremely rapid flowering. We hope that the review can indicate this and we would like to cite. If the reviewer indicated this reference (Woods et al. 2014), we have cited it before.

(From the referee) 9) It is interesting that overexpression of CO1 results in an acceleration of flowering even though VRN2 expression levels increase. Note in barley when CO1 is overexpressed there is a delay in flowering when there is a functional VRN2 (Mulki et al., 2016). These are important results and should be mentioned in the results section of the manuscript and discussed.

Response: As the reviewer suggested, we added these results in the revised manuscript and discussed. Please see the revised discussion section.

(From the referee) 10) Citation 24 and 31 are missing information.

Response: Corrected.

(From the referee) 11) Woods et al., 2014 is cited twice as number 25 and 51

Response: Corrected.

(From the referee) 12) In future, when responding to reviewers comments please include in your response exactly what was added or changed in the main text. At the very least include the line number in the main text where the change occurs.

Response: We marked the modifications and highlighted all changes by in an independent file when we submitted the revised manuscript.

REFERENCES:

- Faure, S., Higgins, J., Turner, A., and Laurie, D.A. (2007). The FLOWERING LOCUS T-like gene family in barley (*Hordeum vulgare*). *Genetics* 176, 599-609.
- Greenup, A., Peacock, W.J., Dennis, E.S., and Trevaskis, B. (2009). The molecular biology of seasonal flowering-responses in *Arabidopsis* and the cereals. *Ann Bot* 103, 1165-1172.
- Halliwell, J., Borrill, P., Gordon, A., Kowalczyk, R., Pagano, M.L., Saccomanno, B., Bentley, A.R., Uauy, C., and Cockram, J. (2016). Systematic Investigation of FLOWERING LOCUS T-Like Poaceae Gene Families Identifies the Short-Day Expressed Flowering Pathway Gene, TaFT3 in Wheat (*Triticum aestivum* L.). *Frontiers in Plant Science* 7.
- Higgins, J.A., Bailey, P.C., and Laurie, D.A. (2010). Comparative Genomics of Flowering Time Pathways Using *Brachypodium distachyon* as a Model for the Temperate Grasses. *PLoS ONE* 5.
- Lv, B., Nitcher, R., Han, X., Wang, S., Ni, F., Li, K., Pearce, S., Wu, J., Dubcovsky, J., and Fu, D. (2014). Characterization of FLOWERING LOCUS T1 (FT1) gene in *Brachypodium* and wheat. *PLoS ONE* 9, e94171.
- Qin, Z., Wu, J., Geng, S., Feng, N., Chen, F., Kong, X., Song, G., Chen, K., Li, A., Mao, L., and Wu, L. (2017). Regulation of FT splicing by an endogenous cue in temperate grasses. *Nat Commun* 8, 14320.
- Ream, T.S., Woods, D.P., Schwartz, C.J., Sanabria, C.P., Mahoy, J.A., Walters, E.M., Kaepler, H.F., and Amasino, R.M. (2014). Interaction of photoperiod and vernalization determines flowering time of *Brachypodium distachyon*. *Plant Physiol* 164, 694-709.
- Shaw, L.M., Lyu, B., Turner, R., Li, C., Chen, F., Han, X., Fu, D., and Dubcovsky, J. (2018). FLOWERING LOCUS T2 (FT2) regulates spike development and fertility in temperate cereals. *J Exp Bot*.
- Wolabu, T.W., Zhang, F., Niu, L., Kalve, S., Bhatnagar-Mathur, P., Muszynski, M.G., and Tadege, M. (2016). Three FLOWERING LOCUS T-like genes function as potential florigens and mediate photoperiod response in sorghum. *New Phytol* 210, 946-959.
- Zikhali, M., Wingen, L.U., Leverington-Waite, M., Specel, S., and Griffiths, S. (2017). The

identification of new candidate genes *Triticum aestivum* FLOWERING LOCUS T3-B1 (TaFT3-B1) and TARGET OF EAT1 (TaTOE1-B1) controlling the short-day photoperiod response in bread wheat. *Plant Cell Environ* 40, 2678-2690.